# EMADDC: high volume, high quality, and timely wind and temperature observations from aircraft surveillance data (Mode-S EHS)

Siebren de Haan[1], Paul de Jong[1,*], Michal Koutek[1,*], Jan Sondij[1], and Lukas Strauss[2]

[1]KNMI, De Bilt, The Netherlands
[2]Austro Control Digital Services, Vienna, Austria
[*]These authors contributed equally to this work.

**Correspondence:** Siebren de Haan (Siebren.de.Haan@knmi.nl)

**Abstract.** Wind and temperature observations from aircraft are of major importance for aviation meteorology and numerical weather prediction (NWP). The European Meteorological Aircraft Derived Data Center (EMADDC) system processes aircraft surveillance data received from air traffic control (ATC) and other partners and converts them into upper air observations of wind and temperature. Only so-called Mode-S Enhanced Surveillance data can be used, because these data contain the air vector and ground vector of the aircraft, from which a wind vector can be inferred. Temperature is derived from true airspeed and Mach number measurements. To produce high quality observations, the data are processed in three steps: pre-processing, processing, and post-processing. The pre-processing is needed to obtain high-quality information and to calculate several correction values for correcting temperature observations and heading values. Processing converts the aircraft data into meteorological information and finally post-processing guarantees that only high-quality information is made available.

The EMADDC system processes around $75 \times 10^6$ surveillance observations per day and produces over $55 \times 10^6$ observations of quality controlled wind observations and $32 \times 10^6$ temperature observations in the European airspace per day. The average age of the observation is around 5 to 10 minutes, depending on the method of data delivery (files via ftp or streaming constantly).

The quality of the observations produced is verified by comparing these observation to other upper air wind and temperature observations from radiosondes and Aircraft Meteorological Data Relay (AMDAR) and comparing them with NWP data. The quality of wind observations is almost identical to AMDAR, the quality of the temperature of EMADDC observations is lower but with bias around zero, while AMDAR exhibits a positive bias of 0.5K.

This paper presents the EMADDC (R2.2) system, operational since 2019.

# 1 Introduction

For normal, and safe, operation, aircraft are equipped with sensors to measure height and velocity with respect to the surrounding air. These sensors can be exploited to observe wind and temperature at the aircraft's location (WMO, 2023). For many years, aircraft observations are an essential component of the global observing system which is used as input for numerical weather prediction models during assimilation (Cardinali et al., 2003; James et al., 2020; Li, 2021; Strajnar et al., 2015; de Haan, 2013). For almost 30 years, aircraft measurements have been collected using the Aircraft Meteorological Data Relay (AMDAR), where meteorological information is automatically sent to national weather services using either satellites or ground stations (Ingleby et al., 2020; Barwell and Lorenc, 1985; Cardinali et al., 2003; James et al., 2020; Lange and Janjić, 2016; Li, 2021; Petersen, 2016; Zhu et al., 2015; Benjamin et al., 2010). Dedicated aircraft are equipped with software to collect the relevant information from the onboard computer systems. Observations are collected with specified observation strategies to optimize coverage with respect to data transmission costs. Over the last decade, a different manner of collecting meteorological information was developed utilizing the operational infrastructure for aircraft safety in Europe, starting in the area of Germany, Belgium and The Netherlands. The infrastructure used by the European Air Traffic Control (ATC) is based on Mode-selective (Mode-S) radars which (selectively) interrogate all aircraft in view of the radar on information on the heading, true airspeed etc., to guide aircraft through its airspace (de Haan, 2011). Although in the whole European airspace Mode-S radars are used for ATC, not all received information can be used to refer to meteorological information, only Mode-S Enhanced Surveillance (Mode-S EHS) radars can interrogate the necessary Broadcast Dependent Surveillance (BDS) 5.0 and 6.0 registers. Fortunately, most Mode-S radars in Europe have EHS capabilities. The observation frequency is determined by the interrogation frequency of the Mode-S radar. Since the information is aimed at ATC, unfortunately no direct meteorological parameters are present in the received information. To extract meteorological information from the received BDS5.0 and BDS6.0 registers, processing and corrections are needed (de Haan, 2011; Stone and Pearce, 2016). Obviously, receiving meteorological parameters directly is preferred above deriving and correcting non-meteorological parameters. Since corrections are inherently imperfect which may lead to complicated errors. The unique character of these Mode-S EHS observations is that (almost) all aircraft are interrogated every 4 to 20 seconds, which results in (locally) very dense datasets.

In light of the COVID-19 pandemic, there was a significant reduction in the number of flights, which in turn impacted the availability of temperature and wind observations collected through the Aircraft Meteorological Data Relay (AMDAR). Nevertheless, as certain airlines continued to operate (e.g. as cargo flights) the European Meteorological Aircraft Derived Data Center (EMADDC) was still producing valuable observations, exploiting the ATC information received for surveillance of all flying aircraft. These observations were used by ECMWF (Ingleby et al., 2021) to address the gap resulting from the lack of AMDAR observations.

This paper describes the current state of the art processing and correction methodology (R2.2) as implemented at EMADDC.

## 2 EMADDC Data Collection

Secondary Surveillance Radar (SSR) is a two-way system where an ATC radar interrogates an aircraft requesting specific parameters. In Europe, all large aircraft (with minimum take-off weight larger than 5700 kg) are required to broadcast Mode-S Elementary Surveillance (ELS) and Enhanced Surveillance (EHS) (European Commission, 2011). EMADDC exploits these to derive wind speed, wind direction and temperature observations from surveillance data requested from aircraft for ATC purposes. Where Elementary Surveillance uses only aircraft broadcasts of altitude and identity, the Enhanced Surveillance interrogation complements these basic parameters with data of the aircraft state, such as roll angle, air speed and Mach number. These additional parameters are requested in groups as a BDS request. To derive wind and temperature, EMADDC requires both BDS5.0 and BDS6.0 to be interrogated and the aircraft to respond.

Additional to these mandatory BDS registers (BDS5.0 and BDS6.0), the BDS4.4 register known as the Meteorological Routine Air Report or MRAR can be also interrogated. This register contains observed temperature, wind, static pressure and humidity (where available). However, this register is not mandatory and only fewer than 5% of aircraft respond to such interrogation requests (Strajnar, 2012) and few countries actively interrogate this register.

### 2.1 Mode-S EHS Interrogation

ATC radar initiates a request to an aircraft for specific BDS data registers. If the aircraft is appropriately equipped, it will respond by broadcasting the requested register information. Each radar employs a distinct interrogation scheme that outlines the frequency of requests and the specific registers to be queried. Different countries or Air Traffic Service Air Navigation Service Providers (ATS ANSPs) may utilize varying interrogation protocols, including different frequencies and rates. The response sent by the aircraft is received by ATC radar but can also be received through a commercially available local receiver, as data is not encrypted. Unfortunately, it is more difficult to decode data received by these receivers as the type of register is not contained in the transmission and hence fuzzy logic or other techniques are applied to decode the type transmitted properly (de Haan, 2011; Stone and Pearce, 2016). An in-house developed C-code software performs this task (similar to the python library developed by Sun (2021)).

### 2.2 Aircraft Dependent Surveillance-Broadcast

Aircraft Dependent Surveillance-Broadcast, or ADS-B, as its name suggests, allows an aircraft to broadcast aircraft state data using the transponder. Data is autonomously broadcast about every 0.5 seconds and contains the aircraft's onboard sensed position (through GPS and inertial systems) which is often more accurate than radar derived position from ATC. This data is available to ATC but can also be displayed on the Navigation Display of newer aircraft for situational awareness. ADS-B does not broadcast wind and temperature, nor does it broadcast all required parameters to derive wind and temperature, although the difference between GNSS height and pressure altitude is transmitted frequently and could be used in data assimilation (Stone and Kitchen, 2015).

## 2.3 Data Handling

As outlined in the previous sections, EMADDC receives data in two ways: 1) aircraft data are collected by ATC, or 2) aircraft data are collected using a local ADS-B/Mode-S receiver. In both cases data is then forwarded to EMADDC through an FTP file transfer. New methods are currently being developed to enhance real-time data transfer, including NewPENS (the pan-European Network Service for real-time exchange of air traffic control data).

Data collected by ATC delivers data of high quality as the content is properly decoded since the content of each transmission is known to the interrogator. However for local receivers, the content of a received register is unknown and logic is required to verify that a register is correctly decoded. All data have quality control and filtering applied.

ATC data can be of ASTERIX CAT48 format, which is mono-radar data, or CAT62 data from a radar tracker combining multiple radars. This latter data format uses filtering to sample all radar plots to a typical 4 second interval. The content of the formats CAT48 and CAT62 is similar but the typical resolution of the Mach number in CAT62 is 0.008 (0.004 for CAT48), and as a consequence the derived temperature are of lower quality (see Section 6). EMADDC is working with EUROCONTROL Maastricht Upper Area Control Centre (MUAC) to develop a solution that provides the Mach number with a resolution of 0.004 and share this solution with other ATC providers.

An advantage of ATC radar data is that the individual BDS register messages corresponding to a single revolution of the radar are combined into a single "observation" message. For ATC radar, the position is determined by ATC radar. In contrast, for data received from local receivers, the position is decoded from the Compact Position Report (CPR) format, which is included in the ADS-B message (and hence not present in radar/tracker data as this data is not available in ASTERIX CAT48 or CAT62 data). The timestamp is not generated by the aircraft; instead, it is created by the radar or receiver at the time of reception. Local receivers utilize either a GPS antenna or a time server for time synchronization. Furthermore, the EMADDC system must combine the various BDS registers to derive observations, as previously outlined.

The techniques utilized by EMADDC are generating significant volumes of high-quality data from Mode-S EHS data. It is essential to implement various quality control that checks identify and address any observation discrepancies and assure that generated observations are of high quality.

## 3 Aircraft Measurement Methodology

A modern aircraft is equipped with sensors that can measure static pressure, Mach number, temperature, position and heading, and Geometric altitude. This section contains a brief description of measurements of pressure, Mach and temperature. The information flow is depicted in Figure 1.

### 3.1 Mach Number and Static and Total Pressure

A crucial measurement in any aircraft is the measurement of the air speed, which can be obtained from the combination of a pitot-tube measurement and a temperature measurement. The pitot-tube measures the static pressure $p_s$ and the total pressure

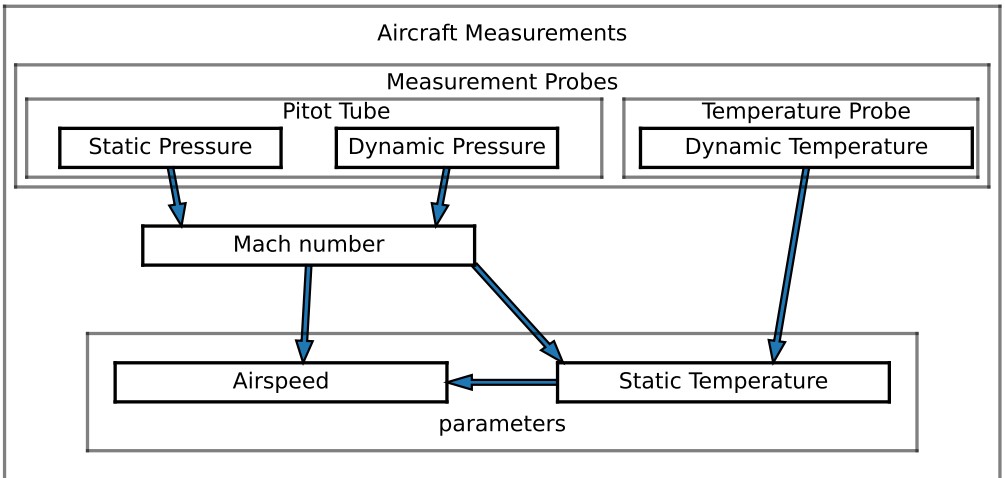

**Figure 1.** Information flow of aircraft measurements.

$p_t$ (Ruijgrok, 1990). Both pressure observations suffer from inaccuracies related to for example a (small) angle between the flow and the probe (Rodi and Leon, 2012). The Mach number is the true airspeed of the aircraft relative to the speed of sound. Let $q_t = p_t - p_s$ be the dynamic pressure, which is more accurately measured because the first order error of $p_t$ and $p_s$ are canceled. The Mach number $M$ is defined as, (Ruijgrok, 1990),

$$M = \sqrt{\frac{2}{\gamma - 1}\left(1 + \frac{q_t}{p_s}\right)^{\frac{\gamma - 1}{\gamma}} - 1}, \tag{1}$$

where $\gamma = c_p/c_v$ is the ratio of specific heats. Note that the dependence of $M$ on the (inaccurate) $p_s$ remains.

### 3.2 Temperature and True Airspeed

The temperature is measured with a temperature probe (Ruijgrok, 1990). The measured total temperature $T_t$ needs to be corrected to obtain the temperature $T$,

$$T = T_t\left(1 + \lambda\frac{(\gamma - 1)}{2}M^2\right)^{-1} \tag{2}$$

The true airspeed $A$ can now be determined neglecting the effect of humidity,

$$A = M\sqrt{\gamma R_d T}, \tag{3}$$

where $R_d$ is the universal gas constant of dry air.

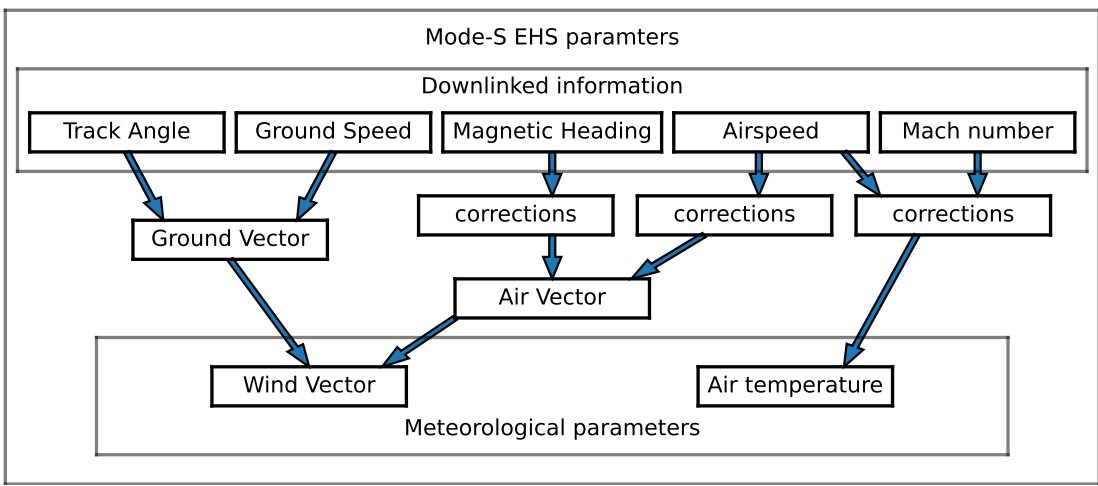

**Figure 2.** Downlinked data flow of Mode-S EHS parameters to acquire meteorological information.

## 4 EMADDC Measurement Methodology

Temperature and wind information is not directly available in Mode-S EHS downlinked information. The wind vector needs to be computed and the temperature needs to be derived, see Figure 2.

### 4.1 Downlinked Parameters

The (most relevant) parameters obtained through interrogation of Mode-S EHS radars are shown in Table 1. The timestamp is created at the moment of reception of the information. All parameters that originate from interrogation, have an observation frequency depending on the radar, ADS-B information can have an observation frequency of twice per second. Table 1 provides information on the downlinked parameters. The information flow of the downlinked parameters is depicted in Figure 2. Also shown in this figure are the corrections applied to the magnetic heading, true airspeed and Mach-number, discussed later.

### 4.2 Raw Data Input Control

The EMADDC quality procedure has been systematically developed and enhanced over the past decade. The first step in defining the quality is to check the input for obvious errors or measurements in conditions where calculation is not possible, as listed in Table 2. Measurements failing one of these checks are discarded from further processing.

### 4.3 Output Control and Whitelisting

Output control is necessary to obtain good quality observations. The parameters for output quality control are related to the correction methods applied to the temperature and wind measurement.

**Table 1.** The reported precision and observation frequency of downlinked parameters, all parameters are rounded.

| parameter | abbreviation | symbol | reported precision | frequency | BDS |
|---|---|---|---|---|---|
| Position (latitude/longitude) | lat,lon | $\lambda, \phi$ | $1 \times 10^{-5}$ deg | 0.5s - 2s | ADS-B |
| Position (latitude/longitude) | lat,lon | $\lambda, \phi$ | $2 - 5 \times 10^{-5}$ deg | 5-20s | radar echo |
| Flight Level | 1 fl=100ft | | 25 ft (7.62 m) | 5s - 20s | ADS-B |
| Roll Angle | ra | | 0.175 deg | 5s - 20s | 5.0 |
| True Track Angle | tta | $t$ | 0.175 deg | 5s - 20s | 5.0 |
| Groundspeed | gspd | $G$ | 2 kt (1.02 m/s) | 5s - 20s | 5.0 |
| Track Angle Rate | tar | | 0.03125 deg/s | 5s - 20s | 5.0 |
| True Airspeed | tas | $A$ | 2 kt (1.02 m/s) | 5s - 20s | 5.0 |
| Magnetic Heading | mhdg | $h_m$ | 0.352 deg | 5s - 20s | 6.0 |
| Indicated Airspeed | ias | $A_I$ | 1 kt (0.51 m/s) | 5s - 20s | 6.0 |
| Mach Number | mach | $M$ | 0.004 | 5s - 20s | 6.0 |

**Table 2.** Current input quality checks used in operation.

| | Input data quality checks | occurrence |
|---|---|---|
| 1 | absolute value of roll angle larger than 2.5 degrees | 16% |
| 2 | absolute difference between track angle and magnetic heading larger than 25 degrees | 1% |
| 4 | true air speed larger than 570 kts or smaller than 100 kts | 1% |
| 5 | groundspeed larger than 850 kts or smaller than 50 kts, or when below flight level 50 smaller than 100kts | 2% |
| 6 | Mach number equal to 0 | 2% |
| 7 | constant flight level and decreasing ground speed and indicated airspeed when flight level is lower than 50 | 2% |
| 8 | position consistency | < 1 % |

Additionally, whitelisting is conducted for wind speed and temperature independently, based on the difference between observations and forecast statistics. Aircraft with a 14-day wind standard deviation exceeding 4 knots are designated as ineligible, while for temperature, the standard deviation threshold is set at 1.23 Kelvin. EMADDC currently uses the operational ECMWF model with a minimal forecast lead time of 9 hours for this comparison by collocating observations with NWP.

## 5   Derived Wind Measurement

The wind vector is the difference between ground vector and air vector, where all vectors are with respect to true North.

$$V \begin{pmatrix} \cos(d) \\ \sin(d) \end{pmatrix} = G \begin{pmatrix} \cos(t) \\ \sin(t) \end{pmatrix} - A \begin{pmatrix} \cos(h) \\ \sin(h) \end{pmatrix}, \tag{4}$$

where $V$ denotes the wind speed with wind direction $d$, $G$ and $t$ are the ground speed and track angle, and $A$ and $h$ the airspeed and heading. Note that this equation is valid under the assumption that the vertical wind speed is negligible, the sideslip is zero and the roll angle is small. The heading is reported with respect to the magnetic North Pole and needs to be converted into a heading with respect to true North. For this purpose, geomagnetic declination tables from (Maus and Macmillan, 2005; Chulliat, 2015) are applied, thus

$$h = h_m + \Delta(y, \lambda, \phi), \tag{5}$$

where $y$ is the datum of the (static) heading correction table on-board the aircraft, and, $(\lambda, \phi)$ is the location of the aircraft and $\Delta$ is the heading correction from the declination table. As it turns out, the heading correction is aircraft dependent, that is $y$ is aircraft dependent, and even may change in time after an aircraft is being serviced, for example when the computer software is updated (Mirza et al., 2016).

## 5.1 Aircraft Dependent Heading Correction

Although the correction should be in the order of the (actual) declination, research showed that a simple correction is not enough (de Haan, 2011; Pourret et al., 2021). Each aircraft may use their own version of a declination lookup table, which implies that each aircraft corrects the true North to magnetic North in a different way. The correction method uses the assumption that the correction is determined by a geomagnetic reference table for a certain datum (or epoch) and is static until updated through aircraft maintenance. The optimal datum is found by minimizing a cost function, depending on datum, by comparing corrected winds from observations with NWP model forecast winds. The cost function is constructed by the vector length difference between the unit heading vector from the aircraft and the unit heading vector formed by the ground vector and NWP wind vector, that is

$$\delta^i(y) = \begin{pmatrix} \cos\left(h_N^i\right) \\ \sin\left(h_N^i\right) \end{pmatrix} - \begin{pmatrix} \cos\left(h_m^i + h_c\left(y, \lambda^i, \phi^i\right)\right) \\ \sin\left(h_m^i + h_c\left(y, \lambda^i, \phi^i\right)\right) \end{pmatrix}, \tag{6}$$

where $y$ is the date of the inclination table, and $i$ the index of an observation, $h_m^i$ is the observed magnetic heading and $h_c(y(\lambda^i, \phi^i))$ the value of the declination table with datum $y$ at location $(\lambda^i, \phi^i)$. NWP heading angle $h_N^i$ is defined as

$$h_N^i = \operatorname{atan}\left(\frac{G^i \sin\left(t^i\right) - V^i \sin\left(d^i\right)}{G^i \cos\left(t^i\right) - V^i \cos\left(d^i\right)}\right), \tag{7}$$

with $G^i$ the ground speed, $t^i$ track angle, $V^i$ wind speed and $d^i$ the wind direction. The first vector on the Equation 6 is the NWP heading direction vector, the second is the corrected heading direction vector.

The cost function is defined as the sum of all vector length differences over all observations, that is

$$C(y) \;=\; \frac{1}{2}\sum_i \|\delta^i(y)\|^2 \tag{8}$$

$$=\; \frac{1}{2}\sum_i \left(\sin\left(h_N^i\right) - \sin\left(h_m^i + h_c^i(y)\right)\right)^2 + \left(\cos\left(h_N^i\right) - \cos\left(h_m^i + h_c^i(y)\right)\right)^2 \tag{9}$$

$$=\; \sum_i 1 - \cos\left(-h_N^i + h_m^i + h_c^i(y)\right) \tag{10}$$

Next, magnetic declination is linearized with datum, in order to find a minimum, that is

$$h_c^i = H_0^i + (y - y_{ref})\Delta H^i, \tag{11}$$

where $H_0^i$ is the value of magnetic declination for given location $(\lambda^i, \phi^i)$ on datum $y_{ref}$, $\Delta H^i$ value of the change in magnetic declination per year (this approximation is valid, as is discussed in Appendix A). The cost function is approximated by a quadratic function in the datum offset, which yields

$$C(y) \approx 1 - \cos\left(H_0^i - h_N^i + h_m^i\right) + \delta_y \sum_i \Delta H^i \sin\left(H_0^i - h_N^i + h_m^i\right) + \frac{1}{2}\delta_y^2 \sum_i \left(\Delta H^i\right)^2 \cos\left(H_0^i - h_N^i + h_m^i\right) \tag{12}$$

where

$$\delta_y = y - y_{ref} \tag{13}$$

The (offset) datum value for which the cost function attains a minimum is found by setting the derivative to $\delta_y$ of the cost function to zero. Let $x_1^i$ and $x_2^i$ for observation $i$ be defined by

$$x_1^i \;=\; -\Delta H^i \sin\left(H_0^i - h_N^i + h_m^i\right) \tag{14}$$

$$x_2^i \;=\; \left(\Delta H^i\right)^2 \cos\left(H_0^i - h_N^i + h_m^i\right), \tag{15}$$

then the datum minimizing the cost function is given by

$$y = y_{ref} + \frac{\sum_i x_1^i}{\sum_i x_2^i} \tag{16}$$

Using the found datum for an aircraft EMADDC calculates the corresponding magnetic declination tables at the location of an observation to find the declination and converts the reported magnetic heading to true heading and calculates the wind according to the equation above.

The above method uses NWP wind forecasts extracted from the operational ECMWF forecast (IFS model). The forecast lead-time is at least 9 hours, such that observation from EMADDC are not used as input for assimilation and correction simultaneously.

## 5.2 Heading Correction Results

Figure 3 shows the results of the heading correction for all 19006 aircraft operational in 2021-2023. Each aircraft is represented by a vertical line in the top panel. The white color indicates that no correction was found and aircraft with a lot of white pixels

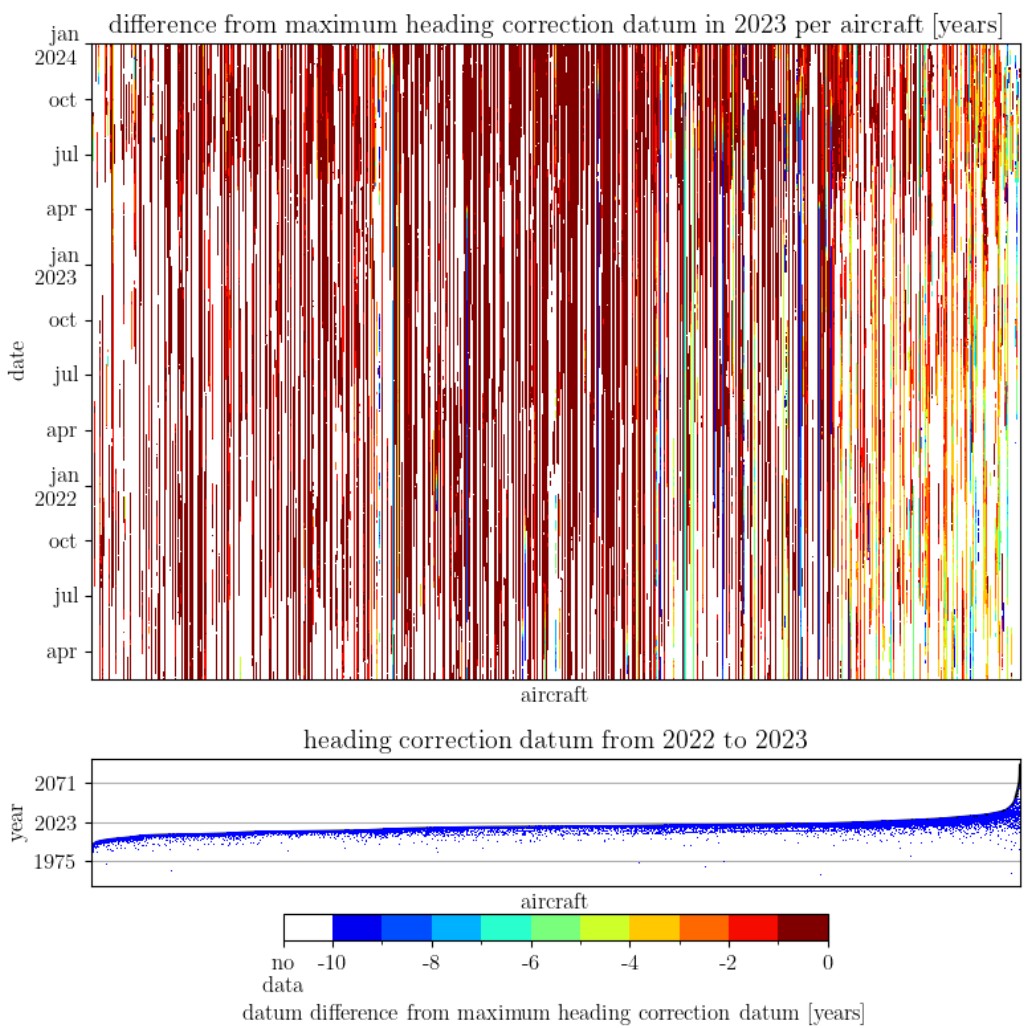

**Figure 3.** Top panel shows the difference of heading correction in 2021-2023 with the reported maximum heading correction datum in this period. The white color indicates that no (or bad) heading correction value was reported. The bottom panel shows the maximum heading correction found in 2023 (top black line) with the deviations from its maximum as small blue dots. The 19006 aircraft are sorted by maximum heading correction.

are not regular flying in the EMADDC domain. The top panel shows the offset with the maximum reported heading correction in 2021-2023. In general, the change in heading correction over 2023 is small, except for aircraft that have high correction datum values. The most constant datum corrections values (offsets smaller than 2 years, red to brown colored) are found for values close to 2023, while higher offsets are only found with high maximum values, which gives reason to believe that for these aircraft the datum correction algorithm may not perform optimally. Note that aircraft near the right of Figure 3 have declination tables decades into the future, and are obviously not correctly estimated, which is also reflected in the noisy pattern for these aircraft. The corrections for these aircraft are invalidated and hence not used.

## 5.3 Dependence on NWP Wind Vector Information

The magnetic heading is calibrated using NWP wind vector information. Consequently, the obtained correction depends on the quality of the NWP information. The magnitude of this dependence is rather small, and of the order of one over the magnitude of ground speed which is explained by the following.

Suppose we have a biased NWP wind direction, that is the true wind direction $d$ is biased by $\beta$, then

$$\tilde{h}_N = \text{atan}\left(\frac{G\sin(t) - V\sin(d+\beta)}{G\cos(t) - V\cos(d+\beta)}\right) \approx h_N + \beta\frac{V(-G\cos(d-t)+V)}{G^2 - 2GV\cos(d-t) + V^2}, \tag{17}$$

which implies that

$$|\tilde{h}_N - h_N| \lessapprox \left|\beta\frac{V(V+G)}{(G-V)^2}\right| \lessapprox \frac{|\beta|}{G} \ll |\beta|. \tag{18}$$

The offset from the true heading using wind direction biased information is substantially smaller than the actual bias.

The difference in heading from an unbiased and a biased NWP model is smaller than the bias divided by the ground speed ($G$). The ground speed is in general around 100 to 200 m/s, and thus the bias in heading is around one hundred times smaller than the bias in the wind vector.

Similarly the offset from the true heading based on wind speed biased (by $\alpha$) is given by

$$|\tilde{h}_N - h_N| \lessapprox \left|\alpha\frac{G\sin(d-t))}{G^2 - 2GV\cos(d-t) + V^2}\right| \lessapprox \frac{|\alpha|}{G} \ll |\alpha|. \tag{19}$$

When the model contains a wind speed bias, the bias in the heading is smaller than the wind speed bias divided by the ground speed, and thus will be a factor of 10 to 100 smaller than the wind speed bias in the model.

Since the heading correction is based on many of observations, over a large period (at least 15 days) it can be regarded as independent of the NWP information. Moreover, the EMADDC system is fitting a declination table over a large domain with a single parameter, the effect of locally biased or erroneous NWP estimates will be small. Aircraft visiting this region will have deviating quotients from which the datum is calculated (see Eq 16). This deviation is detected by checking the standard deviations of the quotients seperately.

## 5.4 True Airspeed Correction

The measurement of true airspeed (TAS) depends on the temperature and Mach number, see Equation (3). Since the observed Mach number is corrected (Rodi and Leon, 2012), the true airspeed measurement might be improved likewise.

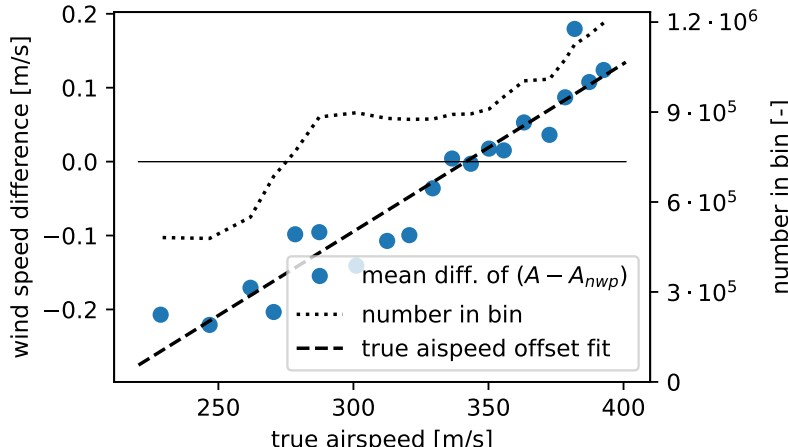

**Figure 4.** The mean difference of true airspeed as derived from observed ground vector and model wind versus the observed true airspeed. Data from three full days (2021/06/05, 2021/12/02 and 2023/08/01) and two 6 hours intervals (2022/08/01 06-12 UTC, and 2023/01/01 00-06 UTC) is shown.

The current algorithm uses an aircraft dependent constant correction value. However, in the next update of the EMADDC processing (EMADDC 3.0), a true airspeed dependent correction will be investigated. This improvement is reasoned by comparing the observed true airspeed with a NWP model equivalent true airspeed as creating a correction that depends on the observed value of true airspeed. The model true airspeed is obtained by the vector difference between the observed ground vector and wind vector from the NWP model. Figure 4 displays the average difference of the observed true airspeed versus the model-based true airspeed. The differences are averaged over true airspeed bins for all data points observed in the EMADDC domain in three days. Note that the difference in wind speed is of the order of a few tenth meter per second. Clearly there is a relation between mean airspeed difference and true airspeed itself.

As a first order approach the EMADDC system currently applies a true airspeed bias correction depending on aircraft and phase of flight. Future research will study physical methods of true airspeed correction.

## 6  Derived Temperature

Although the temperature is measured by the sensors onboard the aircraft, the information is not transmitted in the Mode-S EHS request BDS5.0 and BDS6.0. However, the Mach number $M$ and the true airspeed $A$ are available and from these parameters the temperature can be deduced using the relation between the speed of sound and temperature and the ideal gas law,

$$M = \frac{A}{C}, \tag{20}$$

where $C = \sqrt{\gamma R_d T}$ and $R_d$ is the universal gas constant for dry air. Note that the dependence of the speed of sound on humidity is neglected. So, given $M$ and $A$, the temperature $T$ can be calculated by

$$T = \frac{1}{\gamma R_d} \left( \frac{A}{M} \right)^2, \tag{21}$$

where $A$ is in [m/s].

## 6.1 On Board Aircraft Temperature Correction

The aircraft measurements are improved by algorithms onboard the aircraft. The applied corrections are not available and may be aircraft dependent, or aircraft type dependent, or both. It is known that the measurement of the static pressure $p_s$ suffers from airflow instabilities and/or angle of attack (Rodi and Leon, 2012). The static pressure is corrected, which consequently results in a correction of the Mach number $M$ and temperature $T$.

## 6.2 Temperature Measurement Improvements

The temperature is calculated from the Mach number and the true air speed (TAS) on-board of the aircraft. Actually, there are two types available of the Mach number, one being the downlinked data and one determined using indicated airspeed (IAS) information. The downlinked Mach number is of worse accuracy than the Mach number determined from the indicated airspeed, as will be discussed next. An estimate of the formal error of the temperature $T$ calculated using Equation 21 is constructed following (Taylor, 1997),

$$\sigma_T^2 \approx \left( \frac{\partial T}{\partial A} \right)^2 \sigma_A^2 + \left( \frac{\partial T}{\partial M} \right)^2 \sigma_M^2 = \frac{4T^2}{A^2} \sigma_A^2 + \frac{4T^2}{M^2} \sigma_M^2 \tag{22}$$

Rounding leads to an additional error of $r/\sqrt{12}$, where $r$ is the rounding (see Appendix B ). One of the downlinked parameters is indicated airspeed which is defined as the airspeed measured as if the aircraft was flying at mean sea level ( $p_0 = 1013.25$ hPa, $T_0 = 288.15$, and $\rho_0 = 1.225$kg/m$^3$, the standard pressure, temperature and density at mean sea level)

$$A_I = \sqrt{ \frac{2}{\gamma - 1} \frac{p_0}{\rho_0} \left( \left( \frac{q_t}{p_0} + 1 \right)^{\frac{\gamma-1}{\gamma}} - 1 \right) } \Rightarrow q_t = p_0 \left( \left( \frac{\gamma-1}{2} \frac{\rho_0}{p_0} A_I^2 + 1 \right)^{\frac{\gamma}{\gamma-1}} - 1 \right) \tag{23}$$

The dynamic pressure $q_t$ can be calculated from $A_I$, which in turn can be used to recalculate $M$, according to Equation (1), an (first order) estimate of the error in Mach number becomes,

$$\sigma_M^2 \approx \left( \frac{\partial M}{\partial q_t} \frac{\partial q_t}{\partial A_I} \right)^2 \sigma_{A_I}^2 \tag{24}$$

$$= \frac{A_I^2}{M^2} \frac{\rho_0^2}{p^2} \left( 1 + \frac{q_t}{p} \right)^{-\frac{2}{\gamma}} \left( \frac{A_I^2 \rho_0 (\gamma - 1)}{2 p_0} + 1 \right)^{\frac{2}{\gamma - 1}} \sigma_{A_I}^2 \tag{25}$$

Figure 5 shows the formal temperature errors as calculated with the downlinked Mach number ($\sigma_T^2(M)$, red line) and the temperature error based on the indicated airspeed ($\sigma_T^2(A_I)$, blue line), and the formal error term related to the $A$. The effect

on introducing the Mach indicated airspeed is a reduction in formal error of a factor of 4. The largest part in the formal error is (with the $A_I$) is now related to the true airspeed error. This implies that to further improve the temperature observation reduction of true airspeed error needs to be accomplished.

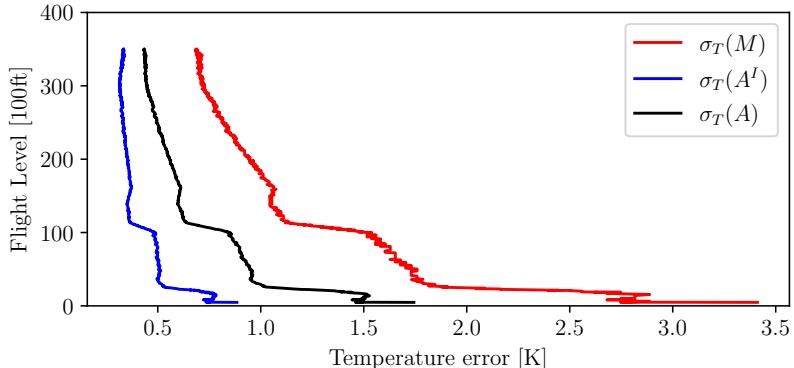

**Figure 5.** Different terms of the formal error for temperature. Black line represents the formal error due to uncertainty in $A$; red line represents the term related to the temperature derived using the (coarse) Mach number; blue line depicts the formal error when the Mach number is recalculated from $A^I$. Aircraft data from ICAO24 A2BD72 (Airbus A320-214), valid from 2023-08-01 05:17:07 to 2023-08-01 05:26:40

The temperature correction is performed in by first averaging and followed by correction. An average over 20 seconds is determined to reduce the noise in the temperature observation, that is

$$\overline{T} = \frac{1}{N} \sum_i^N T_i^I, \tag{26}$$

and average observations are marked as bad when the standard deviation of the observations $T_i^I$ exceeds 5K.

The temperature correction applied is based upon correcting the static pressure measurements and recalculating subsequently
the Mach number, the total temperature and finally the temperature. The correction used is described in detail in de Haan et al. (2022). The corrected static pressure correction $p_c$ depends on the static pressure itself, and the static pressure divided by the square of true airspeed, that is

$$p_c = a + p_s \left( b + \frac{c}{A^2} \right), \tag{27}$$

where the $a, b$ and $c$ are determined by fitting NWP temperatures. These coefficients are aircraft dependent. The corrected
temperature is determined as follows: first the corresponding total temperature is calculated,

$$\overline{T}_t = \left( 1 + \lambda \frac{(\gamma - 1)}{2} M^2 \right) \overline{T}, \tag{28}$$

and then the corrected temperature calculated using the corrected pressure $p_c$ becomes

$$\overline{T}_c = \left( 1 + \lambda \frac{(\gamma - 1)}{2} M_c^2 \right)^{-1} \overline{T}_t, \tag{29}$$

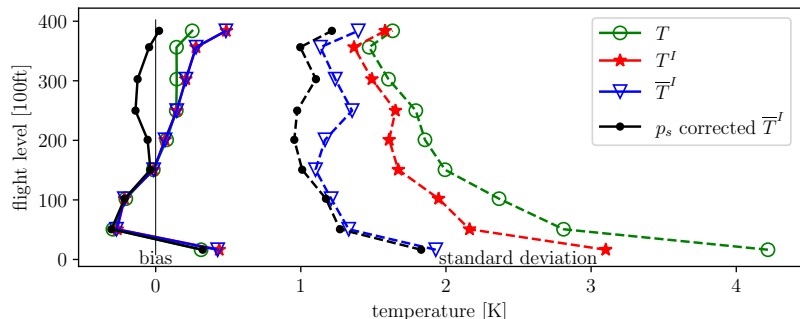

**Figure 6.** Results of temperature correction based on data from 2023-08-01 06-12UTC. The mean difference between model and observation is shown by a solid line and the standard deviation of the difference (dashed line). The green line represents the raw temperature statistics, the red line temperatures calculated using the Mach-IAS relation, the blue line represents additional the time averaging statics, while the black line is the final result after the raw pressure correction.

where

$$M_c^2 = \frac{2}{\gamma - 1}\left(1 + \frac{q_t}{p_c}\right)^{\frac{\gamma}{\gamma-1}} - 1 \tag{30}$$

Figure 6 displays the effect of the three steps (green 'raw' temperature, improving $M$ by the indicated airspeed (red), followed by the averaging (blue) and pressure correction (black) ). Applying a simple smoothing of adjacent points, the standard deviation is reduced to values between 1 [K] in mid-troposphere and growing to 1.5 at higher altitudes and near the surface (Figure 6 compare red and black dashed lines); The bias (and standard deviation to less extend) is then further reduced by applying the raw pressure correction method (Figure 6 black lines).

Figure 7 displays the necessary steps needed in the processing to turn Mode-S EHS observations into meteorological information.

## 7 Data Processing Methodology

EMADDC suppliers generally deliver data in batches every 5 to 15 minutes, allowing the system to retrieve new files for ingestion into the EMADDC database. Receiver data is processed by the decoder, which combines the BDS5.0 and BDS6.0 registers to generate observations that are subsequently entered into the EMADDC database. In contrast, for ATC radar/tracker data, the observations are decoded (from ASTERIX to an EMADDC internal format) and inserted into the database. For this type of data, no intermediate combining step is necessary, as the registers have already been consolidated by the respective tracker or radar system.

Once the data has been ingested into the hourly database table, the processing scheduling system triggers three specific processing jobs. The first job handles observations within a 15-minute time window, operating with an approximate delay of

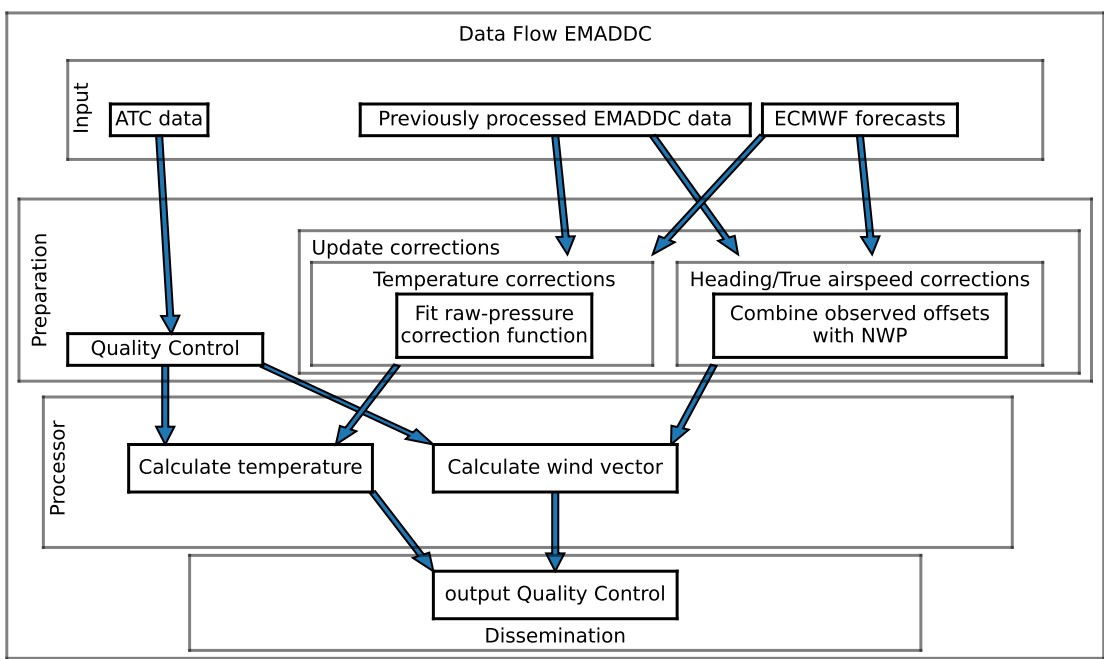

**Figure 7.** Functional data flow in EMADDC, needed to derive wind vector and temperature observations.

30 minutes. This job functions as the primary processing task, indicating that the designated time window has been successfully processed. Figure 8 shows the coverage of EMADDC on April, 21 st 2024.

In 2022, two additional processing jobs were introduced to process observations in 5-minute batches at 13 and 23 minutes past the initial observation in the designated time window. These "fast" files account for approximately 70% and 90% of the total data available in the standard 15-minute interval files, respectively. The implementation of these new fast files has notably enhanced the timeliness of EMADDC by reducing the delivery delay to approximately 10 minutes. These fast files should be utilized when 15-minute interval data is not yet available/processed.

A processing job begins by collecting all available data within the specified time window. The input data undergoes quality control, as previously outlined. To ensure continuity in flight profiles and phase determination, the data from the last five minutes of the preceding time window is included for continuation of flight profiles and phase determination. The flight profile and flight phase are used in the application of the corrections and quality control. Wind and temperature are derived using equations for wind speed and wind direction (see above), the detected magnetic table datum is used in the World Magnetic Model (Chulliat, 2015) to determine the magnetic declination at the location of the observation and obtain the true heading. Subsequent corrections and post-processing are applied, after which outliers are identified using linear regression over individual aircraft's 30-second time windows.

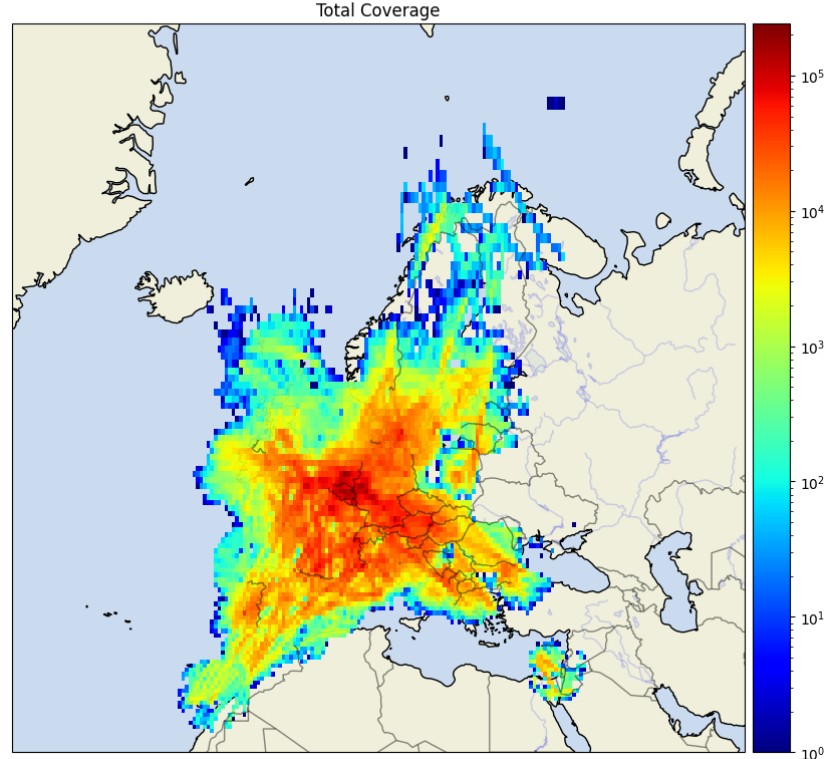

**Figure 8.** Data coverage of EMADDC on April 21st 2024. The color indicates the number of observations per 0.5 degrees squared box.

EMADDC currently receives data from multiple overlapping sources. For instance, in the EUROCONTROL MUAC area, EMADDC acquires radar data from MUAC, in addition to data from approximately four receivers managed by Air Support, which has now been rebranded as ADSB Support. KNMI operates ADSB Support receivers located in De Bilt and Cabauw at
180 meters, which captures data extending to Paris. Since these receivers collect similar data to that of the radars used in ATC, the EMADDC database contains duplicate entries. To address this issue, a duplicate detection algorithm has been implemented, whereby data from the same aircraft recorded within one second of another observation is designated as a duplicate of the primary observation. Observations identified as duplicates are not included in the EMADDC output. It is important to note that observations obtained directly from Air Traffic Control—such as radar or tracker data—are not marked as duplicates, as the
ATC system applies its own filtering and quality control measures to eliminate duplicates. Consequently, it is possible for an observation to appear in a fast file, but be absent in subsequent 23-minute fast files or full time-window files if it was identified as a duplicate and replaced by another observation. The current approach to identifying duplicates has demonstrated effectiveness; however, it is not without imperfections. The intent is to enhance this process in future developments of EMADDC R3.0.

In addition to this step of identifying duplicate observations, the decoding and combining process effectively addresses the potential for duplicate registers that may be received from multiple overlapping receivers within a receiver file. Furthermore, ATC data does not contain duplicate observations, as radars exclusively process received interrogations that pertain to their own interrogations, discarding those produced by other radars.

The final processing step involves verifying whether the observations are from aircraft that have been approved for use through whitelisting as explained in Section 4.3.

Ultimately, all validated, non-duplicate, and quality-checked data are compiled into CSV and BUFR file formats. These files are be made accessible via KNMI's FTP servers and the KNMI Data Platform (KDP)[1]. Access to these historical files or datasets is open whereas operational data is currently restricted to authorized users.

## 8 Quality assessment

Observations derived with the above described processing system are validated with upper air in-situ measurements and numerical model equivalents. This section shows the comparison of Mode-S EHS wind and temperature observations against the ECMWF NWP model, radiosonde wind and temperature observations and AMDAR wind and temperature observations. For the NWP model, the forecast lead time is minimal 9 hours, in order to avoid observation compared that are used in assimilation. The comparison with NWP equivalents is based on three months of data (January - March 2024). Section 8.2 discusses EMADDC versus radiosondes and NWP (January - March 2023). And finally, in Section 8.3 EMADDC, AMDAR and NWP are compared over an eight month period.

### 8.1 Numerical Weather Prediction model Comparison

Wind and temperature observations are corrected using NWP information, as was discussed above. The heading correction applied may introduce some (NWP) correlation but not directly to wind components and temperature and when correlations exists it will be small (see Section 5.3). Furthermore, corrections, as well as whitelisting parameters, are based on historic NWP data, while the comparison is made using operational available (forecast) data.

From January 1st 2024 to April 1st 2024, a total of 4.5 billion observations were derived by the EMADDC system. From these observations 2.8 billion unique and whitelisted wind observations are made available to the users, and in total nearly 1.8 billion temperature observations are disseminated in these three months. The quality of wind observations compared to ECMWF-IFS is around 2.5 [m/s] in standard deviation, with a small bias of 0.3 [m/s], see Table 3. Table 3 also shows the statistics of wind observations for different height levels: the error in wind speed with respect to the model increases with height from 2.2 [m/s] near surface to 2.8 [m/s] at a height of 11 km. The wind direction statistics show a different signal; near the surface the wind direction error is nearly 15 degrees, with a minimum at around flight level 350 and increasing again to an error of 10 degrees at around 11 km. Note that only observations with wind speed larger than 4 m/s are used for wind direction.

---

[1]Please visit https://dataplatform.knmi.nl/dataset/access/emaddc-hist-repro-data-1-0

Wind is in general more variable near the surface. These values are all within the acceptable range for use in for example data assimilation.

**Table 3.** Statistics of EMADDC wind observations against the operational ECMWF model

January 1, 2024 to April,1 2024

| | | | wind speed, EMADDC- NWP | | | wind direction, EMADDC- NWP | | |
|---|---|---|---|---|---|---|---|---|
| | | raw volume | number | bias [m/s] | std.dev [m/s] | number | bias [deg] | std.dev [deg] |
| all data | | 4 546 047 080 | 4 384 070 442 | 0.34 | 4.76 | 4 281 120 981 | 0.14 | 9.62 |
| whitelisted and unique | | - | 2 868 355 459 | 0.30 | 2.52 | 2 800 011 753 | 0.17 | 8.67 |
| flight level | pressure (hPa) | raw volume | number | bias [m/] | std.dev [m/s] | number | bias [deg] | std.dev [deg] |
| 0-100 | 1013 - 696 | 235 608 797 | 151 947 715 | 0.20 | 2.20 | 135 707 126 | -0.16 | 14.24 |
| 100-200 | 696 - 465 | 361 851 780 | 252 516 206 | 0.22 | 2.28 | 243 224 271 | 0.45 | 10.77 |
| 200-300 | 465 - 300 | 594 829 968 | 386 369 304 | 0.27 | 2.53 | 378 271 790 | 0.33 | 9.32 |
| 300-400 | 300 - 187 | 3 133 630 884 | 2 016 820 254 | 0.32 | 2.60 | 1 983 182 045 | 0.12 | 7.94 |
| >400 | <187 | 220 023 701 | 131 427 728 | 0.36 | 2.81 | 128 500 903 | 0.02 | 10.03 |

The temperature statistics are shown in Table 4. The temperature error in total is slightly smaller than 1 [K], with a minimum error of 0.8 [K] around flight level 250. The maximum error is found at cruising level (11 km). Note that the bias with the model is around to zero.

### 8.2 Comparison with Radiosonde Observations

Radiosondes are regarded as anchor observation for meteorology and are generally launched at the main synoptic hours 00 UTC and 12 UTC, with few sites launching also at 06 UTC and 18 UTC. Due to budget restrictions some radiosondes are only launched once a day. Table 5 shows collocated observations statistics of wind (speed, and wind components). Table 6 contains temperature statistics. Aircraft and radiosonde observations will never be exactly collocated in both space and time. Nevertheless, collocations are made by having the maximum distance between aircraft and radiosondes of at most 50 km and

maximum height difference of 100 m and time difference of 1800 seconds.

For all wind parameters, the comparison between radiosonde and EMADDC has mostly a standard deviation lower than that of the comparison model and EMADDC or radiosonde. Furthermore, the difference between model and radiosonde or model and EMADDC are similar and of the order of 0.3 to 0.5 m/s, while the mean difference between aircraft and balloon is small. Reason for this is the model representation error due to the model grid size. The temperature statistics show that all three

systems have the same average temperature (all biases are small and near zero). Not surprisingly, the temperature observations of EMADDC are clearly of less quality than radio soundings, although above 850 hPa the quality is reasonable.

**Table 4.** Statistics of EMADDC temperature observations against the operational ECMWF model

|  |  |  | temperature, EMADDC- NWP | | |
| --- | --- | --- | --- | --- | --- |
| January 2024 - March 2024 | | | | | |
|  |  | raw input | number | bias [K] | std.dev [K] |
| all data | | 4 546 047 080 | 3 138 758 482 | 0.02 | 1.05 |
| whitelisted and unique | | - | 1 763 880 586 | -0.00 | 0.95 |
| flight level | pressure (hPa) | raw volume | number | bias [K] | std.dev [K] |
| 0-100 | 1013 - 696 | 235 608 797 | 51 837 791 | 0.13 | 1.08 |
| 100-200 | 696 - 465 | 361 851 780 | 140 307 524 | 0.05 | 0.83 |
| 200-300 | 465 - 300 | 594 829 968 | 241 388 676 | 0.06 | 0.77 |
| 300-400 | 300 - 187 | 3 133 630 884 | 1 309 098 407 | -0.01 | 1.04 |
| >400 | <187 | 220 023 701 | 83 204 202 | 0.05 | 1.24 |

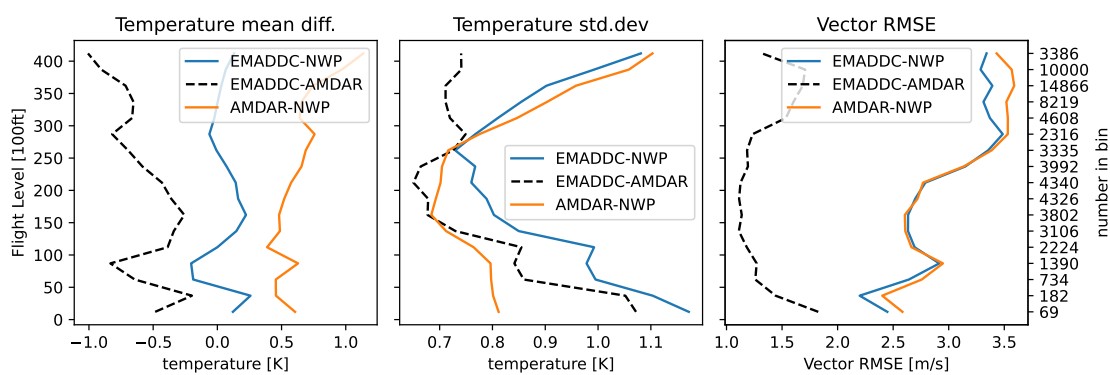

**Figure 9.** Statistics of an eight month period of collocated AMDAR and EMADDC observations. Left panel shows the mean differences in flight bins of 25 FL; middle pane shows the standard deviation of the differences in temperature; the right panel shows the Vector RMS with respect to height [ft].

**Table 5.** Statistics of wind observations comparison against Radiosondes and operational ECMWF model

| Jan 2023 - March 2023 | | | | | | | | |
|---|---|---|---|---|---|---|---|---|
| East-West wind component [m/s] | | | | | | | | |
| | | | EMADDC- RS | | EMADDC- NWP | | RS - NWP | |
| flight level | pressure(hPa) | number | bias [m/s] | std.dev [m/s] | bias [m/s] | std.dev [m/s] | bias [m/s] | std.dev [m/s] |
| < 0 | > 1023 | 166 | 0.58 | 1.16 | 0.40 | 1.41 | -0.09 | 1.08 |
| 0 - 100 | 1013 - 696 | 541 153 | 0.07 | 2.42 | -0.20 | 2.59 | -0.19 | 2.46 |
| 100 - 200 | 696 - 465 | 790 741 | 0.11 | 2.08 | -0.11 | 2.32 | -0.17 | 2.17 |
| 200 - 300 | 465 - 300 | 976 001 | 0.10 | 2.25 | -0.09 | 2.66 | -0.14 | 2.49 |
| 300 - 400 | 300 - 187 | 1 548 851 | 0.05 | 2.36 | -0.06 | 2.69 | -0.11 | 2.55 |
| > 400 | <187 | 54 193 | 0.09 | 2.78 | 0.10 | 2.87 | 0.04 | 2.74 |
| North-South wind component [m/s] | | | | | | | | |
| < 0 | > 1023 | 160 746 | 0.02 | 2.38 | -0.06 | 2.47 | -0.25 | 2.34 |
| 0 - 100 | 1013 - 696 | 434 029 | 0.02 | 2.12 | 0.00 | 2.31 | -0.09 | 2.16 |
| 100 - 200 | 696 - 465 | 565 086 | 0.02 | 2.27 | -0.17 | 2.44 | -0.22 | 2.40 |
| 200 - 300 | 465 - 300 | 858 477 | 0.01 | 2.33 | -0.25 | 2.71 | -0.23 | 2.56 |
| 300 - 400 | 300 - 187 | 590 015 | -0.13 | 2.51 | -0.32 | 2.78 | -0.18 | 2.68 |
| > 400 | <187 | 236 | 0.03 | 1.92 | -0.19 | 2.02 | -0.17 | 1.76 |

| | | Vector RMSE | | |
|---|---|---|---|---|
| | | EMADDC- RS | EMADDC- NWP | RS - NWP |
| | number | VRMSE [m/s] | VRMSE m/s | VRMSE [m/s] |
| < 0 | 166 | 2.18 | 2.06 | 2.05 |
| 0 - 100 | 541 153 | 3.31 | 3.53 | 3.35 |
| 100 - 200 | 790 741 | 2.98 | 3.32 | 3.10 |
| 200 - 300 | 976 001 | 3.30 | 3.81 | 3.66 |
| 300 - 400 | 1 548 851 | 3.40 | 3.89 | 3.70 |
| > 400 | 54 193 | 3.84 | 4.19 | 3.88 |

## 8.3    Comparison with AMDAR Observations

Finally, a comparison is made between EMADDC observation and AMDAR observations. AMDAR observations are extracted from the onboard computer and should have a large resemblance to EMADDC observation and thus the mutual statistics are expected to be small. Figure 9 shows the statistics of EMADDC, AMDAR and NWP for temperature and Vector RMSE. A lookup-table is used to connect AMDAR aircraft to an ICAO aircraft identification. This lookup-table is partially based on E-AMDAR information and is completed with results from collocation with EMADDC observations. Due to rounding of position and time, exact collocations of AMDAR and Mode-S can be tedious. Moreover, the reporting observation frequencies

**Table 6.** Comparison against Radiosonde

| | | | Jan 2023 - March 2023 | | | | | | |
| | | | Temperature [K] | | | | | | |
| | | | EMADDC- RS | | EMADDC- NWP | | RS - NWP | |
| flight level | pressure(hPa) | number | bias [K] | std.dev [K] | bias [K] | std.dev [K] | bias | std.dev [K] |
|---|---|---|---|---|---|---|---|---|
| 0 - 100 | 1013 - 696 | 165 790 | 0.03 | 1.17 | 0.05 | 1.13 | -0.01 | 0.97 |
| 100 - 200 | 696 - 465 | 496 154 | 0.07 | 0.84 | -0.00 | 0.84 | -0.08 | 0.65 |
| 200 - 300 | 465 - 300 | 704 675 | -0.01 | 0.76 | 0.04 | 0.78 | 0.05 | 0.56 |
| 300 - 400 | 300 - 187 | 1 158 841 | 0.01 | 1.04 | -0.01 | 6.47 | -0.06 | 0.87 |
| > 400 | <187 | 36 737 | 0.08 | 1.31 | 0.02 | 1.27 | -0.08 | 1.21 |

are different, resulting in about half of the AMDAR observations being collocated. Here an AMDAR observation is collocated with an EMADDC observation when the time difference is less than 2 minutes, the distance is less than 1 km, and most importantly the height difference is at most 25 ft. The height discrimination is strong, because in general temperature and wind tend to change more with height, than with horizontal displacements.

The temperature of AMDAR has a positive bias, when compared to NWP which is known in literature (Zhu et al., 2015). The bias of EMADDC is around zero. The AMDAR temperature standard deviation is clearly better than EMADDC. The increase of standard deviation near the ground is due to atmospheric turbulence being more present near the surface (Schwartz and Benjamin, 1995). The EMADDC temperature error increases at lower levels because aircraft land generally with a low airspeed and Mach number, which results in a larger error due to the rounding of the airspeed observation given the relatively low resolution of these parameters.

For wind the Vector RMSE shows that both AMDAR and EMADDC perform equally well; there are small differences, but not significant. However, a gross error check on the zonal wind is needed to detect systematic errors in B787 aircraft (WMO, 2022).

## 9 Conclusions and Outlook

This paper presents the current EMADDC system (R2.2) to produce wind and temperature observations derived from Mode-S EHS aircraft messages.

Mode-S EHS is a surveillance method which not only tracks an aircraft in the range of the radar but it also interrogates the aircraft and requests specific information which is used by air traffic services. This downlinked data contains sufficient information to derive wind and temperature at very high spatial and temporal resolution. This data is being processed by EMADDC to produce high quality meteorological information. First of all, to be able to generate observations of good quality, several quality checks are applied. The quality of directly derived wind is hampered by an unknown offset in heading and low resolution of the Mach number and airspeed when deriving temperature.

To obtain high quality wind measurements, a correction from magnetic heading to true heading is necessary; this heading correction is unique for each aircraft and may change in time due to maintenance of the aircraft. The observations are compared to wind forecast of ECMWF model, radiosonde wind observations over a three month period and AMDAR observations over a period of eight months. The derived wind observations compare well to the model and the radiosonde observations with similar statistics as when comparing AMDAR to model equivalents.

The temperature is derived from the quadratic quotient of the true airspeed and the Mach number, with both values truncated. The estimate of the Mach number can be improved by exploiting the downlinked indicated airspeed. Next, a mean temperature is determined using a 20 second time-window, and finally, the temperature is corrected based on the methodology developed in de Haan et al. (2022). Error analysis revealed that the quality of the true airspeed measurement limits the temperature quality. Comparison with radiosonde observations showed good quality with respect to temperature when the observation is above 700 hPa (albeit that the temperature error increase for both AMDAR and EMADDC to a maximum at 200 hPa). AMDAR comparison showed that wind observations from EMADDC are of equal quality, while temperature observations have 25% larger error. The produced meteorological information, when thinned to avoid over-fitting, is at present, widely used in limited and some global numerical weather prediction models.

The advantage of EMADDC over AMDAR is that only few aircraft are AMDAR equipped, and Mode-S EHS is available (almost) everywhere in Europe and therefor all aircraft are being utilized as a sensor. The costs of receiving data are limited, in most area's the data can be obtained through ATC-partners or via local receivers. On the other hand, the AMDAR temperature, although biased, is better for heights below 850 hPa compared to EMADDC derived temperatures.

Although the above described system delivers good quality wind and temperature observations, it is highly preferred to receive the aircraft onboard measurements of available meteorological information directly, as this will reduce and simplify the errors. Current available systems, like the changes introduced in ADS-B Version 3 may be able to provide this information in the future.

A final remark needs to be made on the use of NWP forecast for correction. It is assumed that the NWP model and forecast are (almost) bias free. If this would not be the case then the bias might be reflected with reduced magnitude as a bias in the 'corrected' observations (Eyre, 2016). This paper showed that this is not the case for the heading correction because aircraft heading is not related to forecast values.

## 9.1 Outlook

The EMADDC team tries to improve the quality of derived wind and temperature continuously. The team has the following items to investigate or implement:

– The current heading correction algorithm does not detect changes in the heading table datum effectively, especially when high datum correction values are detected. Revisit of the heading correction algorithm is foreseen in the near future;

– Applying a general true airspeed correction for both wind and temperature;

– Similarly to temperature measurements, a formal error can be derived for wind observations;

- Information on the formal errors are of great interest for users in data assimilation and EMADDC is looking into ways to disseminate this information,

- the vast volume of data per time period and region needs proper treatment for use in for example data assimilation, moreover incorporating formal errors correctly could be accounted for. Research within EMADDC is ongoing on how to apply this most efficiently. Possible methods that will be applied are thinning, and/or super-obbing (i.e. averaging a number of close observations and treat them as one).

- The ADS-B information contains also geodetic height information, which might be valuable for data assimilation. Also information about the aircraft category and positional accuracy is available. The EMADDC team intents to add this information to the observation set created.

- The current system is file based; in near future a (near) real time production is foreseen using the real time networks of ATC in Europe (NewPENS) and real-time transmission of receiver data to EMADDC from ADSB Support and others;

- As part of an initiative of Met Office and KNMI, EMADDC Met Office Global is processing data from Flightradar24 to enhance "global coverage" (limited to aircraft trajectories where Mode-S EHS is actively interrogated). The observations generated through this process are made available as CSV, NETCDF, and BUFR files exclusively to National Meteorological Hydrological Institutes. In time, these observations will be processed within the regular EMADDC system to accomplish synergy for corrections and to prevent data duplication and redundancy.

In the future ADS-B is foreseen to broadcast meteorological information (Rodriguez, 2023) creating enormous data volumes which are expected to require quality control of some kind for use in meteorological applications. The EMADDC system could fulfill the future quality assurance function.

*Data availability.* The processed historical data used in this article is available through the KNMI Data Platform https://dataplatform.knmi.nl.

## Appendix A: Geomagnetic Data

This Appendix shows that the linear approximation in datum is valid for the domain of the current EMADDC processing.

The datum 2015/1/1 was set as the reference datum for the declination table in the current processing setup (R2.2 August 2023). The Geomagnetic model is used in determining the declination for a given position and time (Maus and Macmillan, 2005). Figure A1 shows the value of the declination on 2015/1/1 (left panel), the middle panel shows the yearly change and the right panel shows the difference between declination valid for 2020/1/1 and the linear approximation. The values of magnetic

declination in central Europe are small. The change in declination is strongest for high latitude regions and close to zero for low latitude regions (middle panel). The error made by the linear approximation is small, as can be seen from the right panel.

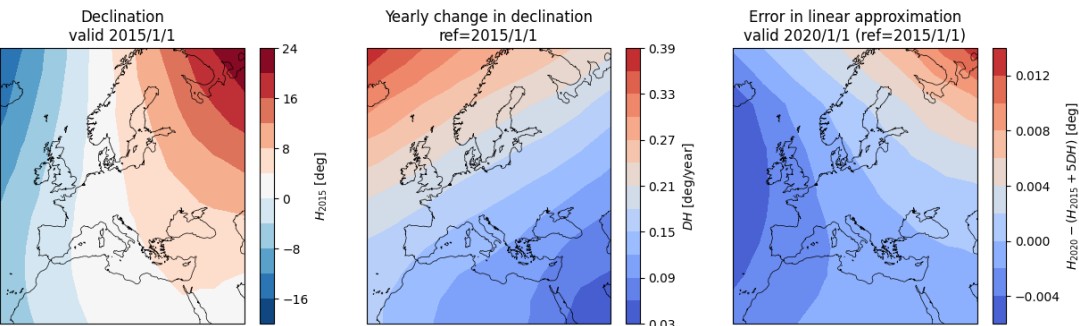

**Figure A1.** The effect of linearization of the declination around the datum 2015. Left panel the declination at 1st Jan. 2015; middle panel the yearly change on 1st Jan. 2015; right panel the difference between linearization and model declination on 1st Jan. 2020. Note that the contours differ for the three panels.

## Appendix B: Rounding Error

The error due to rounding is estimated as follows. Assume that the error of an measurement $X$ is normal distributed with zero mean around the true value $X_t$, and standard deviation $\sigma$. The mean and second moment of the error in $X$ are given by

$$\mathcal{E}(X - X_t) = \int_{-\infty}^{\infty} x \rho_X(x) dx = 0 \tag{B1}$$

$$\mathcal{E}((X - X_t)^2) = \int_{-\infty}^{\infty} x^2 \rho_X(x) dx = \sigma^2. \tag{B2}$$

Let $[X]_r$ be the value of $X$ rounded to the nearest value $r \cdot i$ with $i \in \mathbb{Z}$. The probability of a measurement $[X]_r$ is equal to probability measuring $Y = [X] + R$, with $|R| < \frac{r}{2}$, where the probability density function of $R$ is uniform on the interval $[-\frac{r}{2}, \frac{r}{2}]$. The probability density function of $[X]_r$ is the convolution of the normal distribution of $X$ and uniform distribution of $R$, that is

$$\mathcal{P}(x < [X]_r - X_t) \quad = \quad \int_{-\infty}^{[X]_r - X_t} \int_{-\frac{r}{2}}^{\frac{r}{2}} \rho_X(y-s)\rho_R(s) ds dy \tag{B3}$$

$$\Rightarrow \rho_{[X]_r}(y) \quad = \quad \int_{-\frac{r}{2}}^{\frac{r}{2}} \rho_X(y-s)\rho_R(s) ds dy = \int_{-\frac{r}{2}}^{\frac{r}{2}} \rho_X(y-s)\frac{1}{r} ds dy \tag{B4}$$

thus the mean and variance of error in $[X]_r$ are

$$\mathcal{E}([X]_r - X_t) \quad = \quad \int_{-\infty}^{\infty} \int_{-\frac{r}{2}}^{\frac{r}{2}} y\frac{1}{r}\rho_X(x) ds dx = 0 \tag{B5}$$

$$\mathcal{E}(([X]_r - X_t)^2) - (\mathcal{E}([X]_r - X_t))^2 \quad = \quad \int_{-\infty}^{\infty} \int_{-\frac{r}{2}}^{\frac{r}{2}} (t+s)^2 \rho_X(t)\frac{1}{r} dt dy = \int_{-\infty}^{\infty} t^2 \rho_X(t) dt + \frac{1}{r}\int_{-\frac{r}{2}}^{\frac{r}{2}} s^2 ds = \sigma^2 + \frac{r^2}{12}. \tag{B6}$$

## Appendix C:  Data Sources

Table C1 presents the different sources used in the current processing and Figure C1 shows the coverage of the sources processed in 2024/01. Figure C2 shows the number of daily processed observations since 2016. Clearly visible is the sudden decrease in number of observations during the COVID-19 period.

**Table C1.** Sources of EMADDC in the processing dd. 2024/01

| source | affiliation | main coverage | ATC/local | first data provided |
|--------|-------------|---------------|-----------|---------------------|
| AS-MET | Air Support, Denmark | Europe | local receivers | 2021-04-15 |
| AU | Austro Control | Austria | ATC radar | 2018-09-26 |
| DK | DMI/NAVIAR | Denmark | ATC radar | 2017-11-13 |
| ES | AEMET | Spain | ATC radar | 2019-06-25 |
| FR | Météo France | France | local receivers | 2020-09-08 |
| IL | Israel Meteorological Service | Israel | local receivers | 2023-05-01 |
| MUAC | Maastricht Upper Area Control Centre | Benelux | ATC radar | 2014-01 |
| NO-FFI | MET Norway/FFI | Norway | local receivers | 2021-07-03 |
| RO | ROMATSA | Romania | ATC radar | 2020-10-01 |
| SE | SMHI | Sweden | local receivers | 2021-06-07 |
| SI | ARSO | Slovenia | ATC radar | 2020-09-08 |
| UK | Met Office | United Kingdom | local receivers | 2020-02-01 |

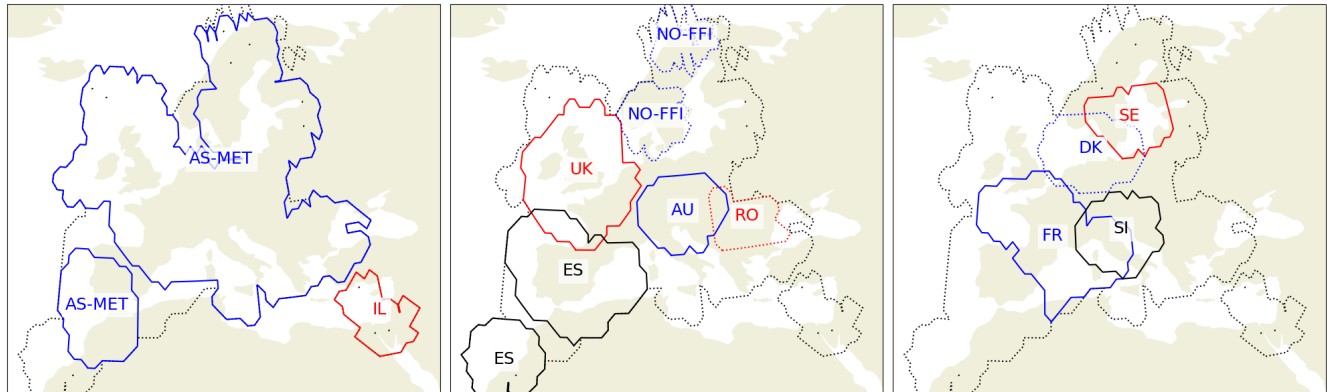

**Figure C1.** Coverage maps of individual sources: left panel sources AS-MET and IL; middle panel ES, UK, NO-FFI, AU; and right panel : FR, DK, SE and SI.

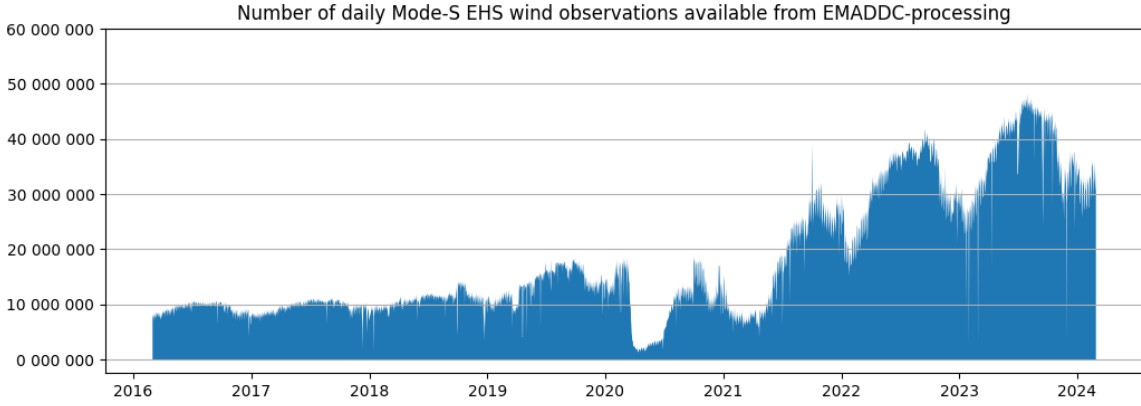

**Figure C2.** The number of observations per day processed by EMADDC over time.

*Author contributions.* Siebren de Haan drafted the manuscript and built the first version of EMADDC and the heading correction algorithm and quality control. Siebren de Haan and Paul de Jong further improved the algorithms and quality control. Paul de Jong and Michal Koutek ported the (research) version to an operational processing system. Jan Sondij is the EMADDC program manager and the liaison between research and the operational aeronautical meteorological service provision and responsible for EMADDC funding and stakeholder management. Lukas Strauss developed the method to improve the Mach number by recalculation using the indicated airspeed. Jan Sondij,

Paul de Jong and Michal Koutek provided input for the manuscript draft.

*Competing interests.* There are no competing interests present.

*Acknowledgements.* The EMADDC system cannot function without input Mode-S EHS data. We gratefully acknowledge the delivery of Mode-S EHS data by all our partners for the generation of meteorological information. In particular the provision of Mode-S EHS data by Air Support (now ADSB Support) and the Met Office / FlightRadar24 via a network of local ADS-B/Mode-S EHS receivers. Special

thanks go out to Torsten Doernbach from EUROCONTROL MUAC who supported the first and important steps with the MUAC data set and provided continuous support. Numerical model data is intensively used for corrections and validation, we therefore thank ECMWF for data provision. The EMADDC program is co-financed (2016-2024) by the Connecting Europe Facility of the European Union.

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
