# Peer review of "EMADDC: high volume, high quality, and timely wind and temperature observations from aircraft surveillance data (Mode-S EHS)"

_Atmospheric Measurement Techniques, 2024_

## Referee Comment (RC1)

Review of the paper "EMADDC: high quality, quickly available and high-volume wind and temperature observations from aircraft using the MODE-S EHS infrastructure "

Dear authors:

Your paper is very nice to read and gives a comprehensive overview of the processing operationally MODE-S EHS temperature and wind data by EMADDC including necessary bias corrections of temperature and aircraft heading and true airspeed. It also shows some information of the MODE-S data quality by comparing the data with a model background and radiosondes. I think the article is well prepared, interesting to read with good readable illustrations and should be published in the Atmospheric Measurements Techniques paper

In the following I have only than minor comments and suggestions

1) In the abstract a short paragraph is missing where you describe what is new in your paper namely the correction of pressure through an improved heading and air speed resulting in a new Mach number resulting in a better temperature.
2) There is no reference of Figure 1 and Figure 2 in the text
3) Equation 1 needs some references
4) In line 49-50 the last sentence is not clear understandable
5) In line 58 correct reseceivers
6) In your description of the Mode-S EHS interrogation it is not clear to me which time stamp the observations get. Is there also a register for that or does the receiving radar gives the observation a time stamp ?
7) I think you can combine Section 2.2 and section 2.3 to get one larger section about Aircraft Dependent Surveillance data.
8) In Section 3 it is not clear to me which way you chose operationally to get better resolved Mach numbers
9) In Section 2.2 you mentioned the possible assimilation of the difference between GNSS height and pressure altitude. Than you assigned a question mark. Is that a speculation of the authors or to you have other information which are ot published ? Can you reformulate that sentence without a question mark.
10) In line 89 correct conains.
11) In table 1 there are two Vertical Rates listed. What is the difference between them ?
12) What is the difference between airspeed and true airspeed ?
13) In section 6.2 you mentioned that a temperature correction is constructed using NWP temperature information but it is never explained further how and at which point you correct temperature by using NWP data.
14) How do you use the corrected pressure further on. Does it replace the ps value in the formula (4) of the Mach number or how do you use Pcor. The whole section 6.2 can be reformulate to make the temperature bias correction clearer.

15) Does the corrected True airspeed also makes the derived temperature via the Mach number better ?

16) In section 8 how do you handle the ADS-B data? In the same way as Mode-S EHS data ?

17) In your result section you talk about flight levels and kilometers or pressure levels. For clarification it would be nice to assign the flight levels also a height in kilometers or a pressure level.

18) The description "The tables below" is unclear. Better to say Table 3 to 6 or so

19) Do you use the ECMWF analyses or a first guess (which) for comparison ?

20) For the wind direction verification do you discard small wind speeds ? This can become important especially near the surface.

21) Are the biases and errors compared to the ECMWF model comparable to AMDAR temperature and wind comparisons? Are the values in table 3 and 4 high or low ? Can you classify the results ?

22) In Table 3 and 4 the units are missing

23) In line 252 Table below again

24) The sentence in line 257 is unclear.

25) Can you further discuss why the Mode-S EHS temperature data are of less quality in the boundary layer beneath 850 hPa

26) The conclusions do not clearly describe the novelty of the work. Also, the comparison results between Mode-S data and NWP forecasts or radiosondes can be described in more detail

---

## Community Comment (CC1)

This paper has the potential to be a valuable contribution to the literature and I enthusiastically welcome the submission.

However, it does leave the reader with some questions which I believe need to be cleared up before submission. Alongside some other smaller corrections and modifications which will make the message of the paper clear.

From my understanding the paper contains two novel themes,

1. EMADDC as a source of high quality aircraft atmospheric observations.
2. An improved method for significantly improved temperatures from Mode-S EHS data.

The other parts are incremental, or explaining how methods previously published have been implemented by EMADDC.

I have two significant concerns with the paper as it currently stands,

1. The conclusions do not adequately explain what is novel in the paper. If you were to read it without the context of the entire paper you would just write it off as not novel. The story needs to be made clearer throughout with the conclusions rewritten to draw out the importance and novelty of this work.
2. I believe there is a processing method which is mentioned but not adequately explained. The reported quality of the temperatures is significantly greater than that reported previously. It is suggested that this is via creating a corrected pressure and recreating the Mach number. This is never explicitly stated and the method to do this is in no way explained. It is hinted at several times in the paper e.g. Line 80 "derive the Mach number from indicated airspeed", Table 2 (6) the stated minimum Mach is below the resolution limit of the reported Mach and CCC – for CCC the contents of the personal correspondence needs to be detailed, or a further reference found. This is fundamentally the crux and key novel development for Mode-S EHS processing. One of the problems seems to be that the novelty is contained in Straus 2020 which is a personal communication, this needs significant expansion.

I have some less significant comments,

3. The abstract does not fully convey the novelty of the paper, it should also include some highlights of the results e.g. improvement of temperature observations.
4. The use of 'Airspeed' throughout to mean (probably) True Airspeed, this should be stated explicitly at the first introduction or 'true airspeed' should be used throughout, else it is ambiguous with Indicated Airspeed, True Airspeed, Calibrated Airspeed or Equivalent Airspeed.
5. You use altitudes, flight levels and pressure levels in hPa. Please choose one and use it consistently, e.g. Line 262 refers to 858 hPa when refereeing to table 5, table 5 only has flight levels.
6. Section 2.3, this seems superfluous to the rest of the paper.

7. Line 81, where ATC combine the messages to create an observation, how is the time stamp of that 'observation' obtained? The aircraft never reports a time. This is followed up on Line 83 where you talk about it for the receiver, but surely the radar must work the same?

8. Table 1 should also include the reported resolution of the variables.

9. Table 1 it is unclear why there are two "Vertical Rates" in the third section and what each section refers to. It would also be good if this table included the reporting resolution. The BDS table refers to BDS numbers which are not discussed in the text, that makes it unclear, especially as I think some of them are ADS-B messages?

10. Table 6 needs column titles.

11. Lines 140 – 143, You need to explain how Pcor is used to make a new Mach number and/or Temperature? This appears to be phenomenologically derived, is there any physical or instrumental reason for it? I think you need to recreate some of the AMDAR plots for Mode-S to demonstrate it's also true. Can this method really deal with the spread at low M values (e.g. when the aircraft are flying slowly).

12. Table 3, have you considered using the components of the wind rather than speed and direction? That may explain the larger wind direction standard deviations at lower altitudes where the winds tend to be lighter.

And some more minor comments,

13. Figures 1 and 2 appear to not be referenced within the text.

14. Line 22 "The last decade…" – the sentence does not make sense.

15. Line 31, I would also include https://doi.org/10.1175/JTECH-D-15-0184.1 in the reference.

16. Line 40, remove "so called"

17. Line 41, the reference being a number makes it unclear.

18. Line 46, add "and broadcast" to the end, as the broadcast messages are also required.

19. Line 48, add "(where available)" after humidity.

20. Line 57, first use of ADS-B.

21. Line 58, the sentence from "is not contained…." Is unclear, you should also reference https://doi.org/10.1175/JTECH-D-15-0184.1 and https://doi.org/10.1029/2010JD015264

22. Line 66, replace "could be used in DA" with "contains atmospheric information" or similar and reference https://doi.org/10.1175/JTECH-D-14-00192.1

23. Line 77, what is the typical resolution of the Mach number in CAT62?

24. Lines 85 and 86 need rewording as they're unclear.

25. Line 88, your wording of "static pressure or pressure altitude" is somewhat difficult, the pressure altitude is never measured but calculated from the static pressure.

26. Eqn 1, requires referencing.

27. The use of both static and ambient temperature, whilst the meaning is consistent it is probably worth picking one to use consistently, potentially highlighting that it also means the other once in the paper. You may also want to state that this is what NWP/Forecasters expect.

28. Line 105, "arrival of information" – it is unclear as to where this arrives.

29. Line 116, over what time period was the standard deviation calculated?

30. Table 2, how can '6 Mach number smaller than 0.001' be true when the reporting resolution of Mach number is 0.04?
31. Line 132, does it not also result in a correction to the Airspeed?
32. Line 135, what evidence is there that pressure is less accurate at low altitudes? This needs to be referenced, or explained and evidenced.
33. L148 the WMM you are using is quite old, why? My assumption is that your heading corrections will deal with that. IIRC the Met Office uses the IGRF which tries to model the current fields and is updated every year.
34. Line 152, there's a Mirza paper that you should reference. https://doi.org/10.1002/qj.2864
35. Line 185-187, I'm a little confused, do you or do you not use the WMM tables? I'd be really interested to see o-b maps per aircraft.
36. Line 204, could you estimate the order and size of each of the corrections, and reference to the current method of correction?
37. Line 208, are the ADS-B data decoded and stored in the same way where it's available?
38. Line 215, do you no longer do any interpolation between data points? You don't mention it here but it seemed like it was a significant part of your earlier papers processing methods and based on some recent data discussions it appeared you were still doing it.
39. Line 219, that's interesting.
40. Lines 226-229, Have you compared these? Primary radar position is worse than ADS-B so why not do something clever in combining?
41. Line 234, "the tables below" which? Reference them, or edit this introduction sentence as it's a bit confusing.
42. Line 237, replace collected with derived as I suspect many more messages were collected!
43. Lines 240 – 244, is this good, bad or indifferent? How do those values compare to other measurement methods?
44. Line 247, do you have any explanations for this?
45. Line 252, reference precisely which table.
46. Table 5, is this only the matched observations for EHS – NWP, you should be clear on this point.
47. Line 256, what are the average coherence lengths of wind and temperature fields for these?
48. Table 6 needs column titles.
49. Kube 284, "middle panel" doesn't seem to agree with the figure, unless I'm misunderstanding and therefore could you edit the words to make it clearer?

---

## Community Comment (CC2)

Review of: EMADDC: high quality, quickly available and high-volume wind and temperature observations from aircraft using Mode-S EHS infrastructure, by de Haan, et al.

Summary Statement: Let me begin by saying that I am a strong supporter of the use of Air Borne Observations (ABOs) in all phases of operational forecast, from bench forecasters using the data to improve short-range local forecasts and hazardous weather warnings to integration into NWP systems. Many studies have shown the impacts of ABO data in both applications, especially in otherwise data-poor areas or between conventional radiosonde launch times. Aircraft position and movement information included in Mode-S reports could provide a possible alternative to the direct measures of temperature and wind (and in some cases moisture) provided by more established AMDAR program. A major disadvantage of the AMDAR is the cost of receiving the data via air-to-ground communications in some regions of the globe, while a major disadvantage of relying on Mode-S is that meteorological information must be derived from position and aircraft motion information provided in reports that were originally designed for air traffic management purposes. This paper, along with others by the lead author, takes the position that the risks and possible errors in the deriving larger volumes of meteorological parameters override the costs involved in acquiring more directly observed AMDAR data.

Although I found the section of the paper describing the need to correct for the difference between true north and magnetic north to be thorough and well presented, I found that other parts of the paper, especially those related to determining temperature, to be vague, overly optimistic and not only reproducible, but also possibly incorrect for applications in areas of the atmosphere with significant moisture. Some of these same issues appeared in earlier papers referenced in this submission. Overall, I must recommend that the paper be returned to the authors so that they can make major revisions before reconsideration for publication.

Specific Comments: As I read the paper, I kept thinking that the authors might have submitted an earlier version than intended. At the end of line 66, there is a '(?)' in the end of the sentence. What does that mean? The sentence is also conjecture and probably should be eliminated. Throughout the text, there are many instances where statistics that could be quantified are instead replaced by vague adverbs of adjectives, such as the word 'frequently' in the same line. Similarly, in line 53, the words 'not all' should be quantified. As it stands, it could mean that as few as 1% or well over 50% of radars would not meet one of the 2 conditions described in the sentence. Also, Table columns are incorrectly labelled and variables in some of the equations are not clearly defined. Numerous spelling and grammar errors also need to be corrected throughout the paper.

Lines 56-59: With the large volumes of Mode-S observations available, how much information does the inclusion of questionably encoded reports add to the volume of reports from more reliable transmissions? Please indicate how much these reports might degrade the overall quality of the derived data sets.

Line 60:  It would be very useful to list the transmitted parameters that are more important in deriving each of the meteorological parameters early in the paper.  E.g., it would be helpful for the reader to know ahead of time which observed parameters affect temperature derivations.

Lines 73-86:  This section describes at least 3 different means in which Mach number that are used at EMADDC.  Please explain to the end user how they can know which of the three options were used for in deriving meteorological data from each aircraft and how that choice might affect the quality of the reports and how much difference each of the 3 methods makes.

Line 89:  Do the Mode-S reports include GPS horizontal position reports as well as altitude?  This sentence implies that they do not.

Equation 1: It would seem more logical to identify the dynamic pressure at $p_d$ instead of $q_t$. Also, the variable in equations 1 and 2 need to be defined in the text.

Section 4.2:  Since this is not relevant to Mode-S observations, this section is unnecessary.  Also, the equation, if used, should be written so it is solved as T=, not $T_i$=.

Table 1:  This table is incomplete and incorrect in places.  Frequency and units are missing for position, even though an accuracy was given in their 2022 paper.  The labels of the frequency and reporting accuracy columns are also reversed.  Also, although time was listed as a coded parameter in the authors 2022 paper, it is not listed here, nor are the precision of the reported value.  This needs to be clarified, since the 2022 paper lists a choice of 2 values (1 s or 1 ms, where 'ms' is undefined).  Which is used in your current system?  If both are used, what impact does that difference have on derived meteorological variables?

Also, no discussion is presented in this or previous papers about how the onboard reports are 'binned' into their reporting precision intervals.  Specifically, were the reports simply truncated was software included to determine if the reports were within +/- ½ of the precision interval on either side of the reported value.  This information is essential to determine if biases have been introduced in the data compression process.

Lines 104-109:  Nowhere in the paper are the common frequencies used for the various parameters used in deriving meteorological information specified.  This is especially hard for a reader to guess since the frequency is listed as a range.  In their 2011 paper, the authors indicated that a 15 (or 60) second averaging (or linear fit) of Mach number and air speed was necessary to improve derived temperature calculations.  That statement is not repeated in this paper.   Has this changed?  If so, say so and explain why.  If averaging is used as part of the calculations, then only 1 derived parameter should be reported during the entire averaging period to avoid correlated errors between successive corrections and the reporting frequency should be adjusted to reflect that change.  This need to be clarified and well documented.  Also, please show which of the two parameters (Mach number or air speed) benefited more for using the linear fit smoothing process?  Also, and probably most importantly, is the question of whether the corrections applied to both parameters in the linear smoothing process are

correlated or uncorrelated in instances where the method improved the derived temperatures. (For reference, investigation of Mode-S wind speed that I have done using a small random sample of data provided ECMWF shows observation-to-observations wind speed changes frequently approaching +/-2 m/s between successive 20 second reports, even after applying a 3 sigma QC filter. This variability could have major consequences on the quality of more instantaneous temperature derivations.)

Lines 115-118: Having looked carefully at a substantial amount of Mode-S derived meteorological reports, I recommend that error bounds of 2 standard deviations be used instead of 3. This more conservative approach is especially justified give the large volume of Mode-S reports and will reduce the data volume by no more than a few %.

Table 2: Please describe how and why these limits were chosen, especially for test 7. Also, if Mach number reporting accuracy is .004, why is .001 used in test 6.

Lines 120-143: This section of the paper concerns me most for numbers of reasons. First, and possibly most significant, is that the fact that the speed of sound (and therefore Mach number) is affected by atmospheric moisture. The 2022 paper explicitly states that the effects of moisture are ignored. Although the effect of moisture is indeed small at upper cruise levels, impacts in the bottom several hundred hPa can be significant, increasing the speed of sound in the moist, less dense environments by over 2 m/s. This can in turn affect temperature derivations by as much as a degree and would result in the derived temperatures being more like virtual temperature than a sensible temperature. This system shortcoming must be recognized. In addition, this large of a change in reported Mach number could shift the transmitted Mach number by one or two of the 0.004 reporting precision increments, which could lead to errors in reported Mach number of up to .008, which could impact temperature calculations even further.

Line 129-132: As with all other corrections applied in this paper, please give a typical magnitude and range of values for these corrections. In this case, how much does the static pressure correction affect both Mach number and subsequent derived temperatures?

Lines 134-143: This paragraph is quite confusing. The first sentence states that NWP is being used to correct temperatures, but no explanation is made of what NWP information is used or how it affects the correction. It implies that corrections are needed for each aircraft individually, but no explanation of the reason for this is documented. A correction formula (formula 5) is then presented based on a new set of static pressures apparently derived from NWP fields. The 2022 paper is then references for more details, but I could not find any there. Instead, the paper describes a method of correcting AMDAR temperature observations, not Mode-S temperature derivations. An explanation of the derivation of the coefficients in (5) is essential for this technique to be reproduced by others, along with examples of the magnitudes of the correction that are applied as a function of altitude. As a side point, one possible explanation of the effectiveness of the technique is that, since the NWP height fields on

pressure surfaces are derived using virtual temperatures, this correction could be unknowingly accounting for the effects of water vapor. This needs much further discussion.

Lines 201-205: Although the section on correction for true vs. magnetic north is detailed and seems sound, the statement in lines 201-205 not well documented. No printed reference is given for the 'true air speed bias correction' that EMADDC uses, including no indication of the magnitude of the corrections. It also seems to assume that most of the wind errors are in speed and that wind directions cannot be corrected. Is this true? If so, it should be stated directly. Again, without information about how this correction was formulated, the work cannot be duplicated by others. Finally, if a future physical correction method depends on an already corrected temperature, how might the 2 corrections interact?

Section 9.1-9.2: Line 239 refers to the quality of Mode-S wind data using parameters u, v and wind speed. This list, however, fails to include possibly the best measurement of overall wind observations quality, that being the Vector RMS (VRMS), which accounts for both wind speed and direction errors. Although wind speed fits were similar using both NWP and Radiosondes as comparison standards, approximations of the Vector RMS derived from the u and v fits with radiosondes produce VRMS values closer to 3.25 m/s. Comparisons of reports between AMDAR and radiosonde data over the US show wind speed fits less than 2.0 m/s and VRMS values of about 2.5 m/s throughout the depth of the troposphere. TAMDAR reports were substantially worse, with speed fits ranging from 3-4 m/s and VRMS values of 4.5-5.5 m/s. I recommend that you expand your references to include the US intercomparisons, expand your statistics to include VRMS and make specific reference to the lower quality of the individual Mode-S reports when compared to previous AMDAR evaluations. That said, I believe that if the Mode-S reports were amalgamated over periods of 5-7 minutes (the typical time between AMDAR flight level reports), much of the small-scale noise that I have observed will be removed the statistics would improve substantially. Please make the labeling of column headers in Tables 3-6 consistent, more clearly explain the meaning of 'all data' and 'whitelisted and unique' in the caption. Also, please make the layers in Tables 3 and 4 and in Tables 5 and 6 consistent so that the 2 sets of results can be compared directly.

I was happy to see that the authors used both NWP and radiosondes in their evaluations. As stated earlier, reliable reports taken by individual aircraft during ascent and descent could be extremely useful for operational bench forecasts, especially in land areas without radiosonde coverage and impending hazardous weather.

Finally, although the authors have gone through great lengths to in efforts to derive temperature information from the Mach number of air speed observations available through Mode-S, many fewer temperature derivations were made than the number of wind reports that were made. This reasons for these differences need to be explained (and understood) more clearly in the text, including the methods by which nearly 35% of the derived temperatures were rejected.

With this substantial number of questions remaining to be answered, I can only recommend that the paper be returned to the author for major revision and resubmission.  I encourage the author to do so.

---

## Author Comment (AC1)

Dear Reviewer,

We thank you for your comments and suggustions to improve the manuscript. We have taken the oppurtunity to add several things, some wanted by you some by other reviewers. The main changes in the manuscript are:

5
- We changed the title to "**EMADDC: high volume, high quality, and timely wind and temperature observations from aircraft surveillance data (Mode-S EHS)**

- We comparison with AMDAR data to the results section

- We included Vector RMS statistics in the results section

- We explain in more detail how the seperate steps for temperature corrections

- We revised the conclusion and added an outlook section

10
- Lukas Strauss has been added as co-author to give him the credits for inventing the Mach-Indicated Airspeed improvement.

We hope to have answered all remaining questions as good as possible.
Thank you sincerly for your time and effort, Best regards, Siebren de Haan and Co-authors

**Answers**

15  Given the importance that observations derived from MODE-S EHS data are taking on, both for operational meteorology and for research, and the amount of algorithm development and data processing techniques that have been necessary to achieve the ability to reliably produce so much useful data, the publication of such an article is fully justified.

The article follows a logical order, provides a synthesis between work already described, which is referenced and placed in the context of the present work, and more recent aspects, such as aircraft-dependent heading correction (§7.1). It ends with

20  insightful characterisation of the data produced versus NWP and radiosondes.

Nevertheless:

the choice of journal remains, in my views, open to question. For example, could Earth System Science Data (ESSD) be a more appropriate choice? In its current form, the article needs to be corrected or reworked before it can become a solid reference for all future work using these data, which it ultimately deserves.

25  In several places, the article lacks precision of description, and relies on the reader's implicit understanding. This needs to be corrected in a definitive publication. In particular:

ok  in §8 Processing Infrastructure, the text let the reader think that the duplicate removal process applies to calculated observations, and not to input (Mode-S and ADS-B) messages. This would means that each data supply channel is treated individually, and that the duplicate removal process is applied at the end, when all the groups of observations produced are merged. Is this really the case? As the article states that "A processing job starts

30  by gathering all data available in the time window of interest." (line 212), it seems that there is room for de-duplication of input data (for example, same MODE-S message received by 2 receivers). Is this carried out, or not ? This should be clarified.

ok  are the whitelistings described in §5.3 "Output control" and whitelisting at the end of §8 "Processing Infrastructure" (around line 230) the same? If so, it could be better described in one place, and simply referred to in the other one ("observations are within three times standard deviation of the measurement with NWP model equivalents" does not make much sense to me).

35  ok  Also : some processing techniques depend on assumptions (for example, magnetic declination tables or the form of corrections for static pressure, Mach number or airspeed). Overall, the final results on the quality of the measurements produced validate the work carried out and the assumptions made, but for this article to give the reader a full understanding of the measurement and processing techniques, quantified indicators should be given for the various stages. For example: are there any aircraft for which minimisation of the cost function for magnetic declination (eq. 18) does not converge? Is so, do the authors have any clues about these aircraft (particularly old or recent, or else)? What is the typical amplitude of the true air

40  speed correction mentioned in §7.3?

*we added a figure on this... around at most a few m/s* In addition to the ongoing research mentioned to develop a more physical method, did the authors try to check that this correction was indeed uncorrelated with a spatial characteristic, or some bias in the model, or else (for example, simply by drawing maps of typical values, or scatter plots, . . . ) ? What is the typical percentage of aircraft that are whitelisted? Is it evolving over time ? Is it possible to learn anything from the list of aircraft that are rejected ? Are they simply aircraft that transmit incorrect data, or are they particular types of aircraft for which other assumptions and calculation methods could be used? This could be an avenue for development if these aircraft fly where others do not.

*we hope to answer these questions thoroughly by the foreseen research*

Here are some more specific remarks, along the text :

ok Line 15 : "For many years, aircraft observations form the backbone of the global observing system". The wording "form the backbone" appears a bit overstated. Associated references support the value and importance of aircraft observation, but they do not assess with certainty that it is central and structuring. Could be rephrased as "aircraft observations are an essential component of the global observation system"

ok Line 17: De Haan, 2013 is an article which deals with Assimilation of GNSS ZTD and radar radial velocity for the benefit of very-short-range regional weather forecast. It recalls the importance of aircraft-based observation, but only marginally demonstrate it. I don't think that its brings much here.

ok Line 33: Since there is a causal relationship, the wording 'However, as (or since) some airlines have continued to fly' seems more appropriate to me than 'However, whilst some airlines have continued to fly' (but I'm not a native English speaker).

ok Line 58-59 : You might want to quote a technical article on these decoding techniques, to give the interested reader the information they need to find out more about these difficulties and the techniques for overcoming them. For example : J. Sun, H. Vû, J. Ellerbroek and J. M. Hoekstra, "pyModeS: Decoding Mode-S Surveillance Data for Open Air Transportation Research," in IEEE Transactions on Intelligent Transportation Systems, vol. 21, no. 7, pp. 2777-2786, July 2020, doi: 10.1109/TITS.2019.2914770.

ok Line 66: missing reference at the end on the sentence "[. . . ] transmitted frequently and could be used in data assimilation (?)." . For example, Bruce Ingleby mentioned such a technique in is poster at the 2023 International Symposium on Data Assimilation (ISDA-2023), titled "ECMWF use of Mode-S winds and changes to aircraft thinning."

ok Line 80: Since reference is made to a personal communication, details of the calculation should be given in the present text.

ok Line 83 : "the timestamp is supplied by the receiver and not by the aircraft" is not the root cause of the difference between receiver and radar data, since "The timestamp is created at the moment of arrival of the information", as stated in §5.1. The advantage of the radars probably rather comes from the synchronisation between the positioning, carried out when the echo from the aircraft is received, and the reception of the Mode-S message, practically simultaneously

ok Line 95, eq.1 : it could be worth noting that the numerical constant used here are valid for dry air, and later, for example in in §7.3, consider the possibility of controlling the applied correction in areas known to be particularly humid (boundary layer in the Mediterranean or the Canaries)

ok Line 104 : given the prior presentation of the difference between receiver and radar data, and even if I agree with the need to choose "The (most relevant) parameters", I would expect table 1 to contain two different lines for positioning : (latitude-longitude) from ADS-B at 0.5-2 seconds period, and (range-azimuth) from radar at 5s - 20s period. Or EMADCC never uses radar positioning and mixes radar and ADS-B receiver to assign a position to an observation ? Also, in table 1, the headings of the 'frequency' and 'reported accuracy' columns are reversed.

ok Line 112: "check the input for obvious errors," could be completed by "or measurements in conditions where calculation is not possible".

ok Line 146, eq.6 : Even if their meanings can be guessed, I believe that "V" and "d" have not been formally defined before, and it would be more rigorous to do so. Please also recall quickly the hypothesis behind this formula (this is a 2D formula, not valid at large values of roll and pitch angles, which justifies the criteria roll <2.5% in table 2, and this formula assumes that the airspeed is aligned with the axis of the aircraft (the heading), and therefore that sideslip is zero, which is mostly true for airliners, but not necessarily during the aircraft's rapid manoeuvring phases)

ok Line 207 : after the sentence beginning with "For receiver data. . .", the reader expects another one describing what is done for radar/tracker data. Or does this sentence apply to both receiver and radar data ?

ok Line 241 and further ; Although I know that this is common practice, the choice of the word 'error' to designate the difference between an observation and a model analysis is, to say the least, debatable. There are cases where the RMS time series of these deviations have changed significantly, without any change in the observation system, but when the model version was updated. Especially since the article later shows that "the comparison between radiosonde and Mode-S EHS show to have a standard deviation lower than that of the comparison is model and Mode-S EHS or radiosonde", which suggests that part of the variance in the MODE-S/model difference is due to a discrepancy between the model and the reality.

ok Figure 4 legend : an "i" is missing in "Observatons"

ok Line 262 : "858" is probably a typo for "850"

ok Line 271 : I don't fully understand the sentence "Note that, although the data is corrected using ECMWF forecast, the data is independent because a forecast lead time of minimal 9 hours is used". Is the forecast lead time of 9 hours used for the computation of the magnetic declination table, or the True airspeed correction mentioned in §7.3? Wouldn't this sentence be better placed closer to the correction description ? Does it imply that the impact of Mode-S assimilation in the model forecast does not extend beyond 9 hours?
*Added words in heading correction section "Derived wind measurements"*

---

## Author Comment (AC2)

Dear Reviewer,

We thank you for your comments and suggustions to improve the manuscript. We have taken the oppurtunity to add several things, some wanted by you some by other reviewers. The main changes in the manuscript are:

– we changed the title to "**EMADDC: high volume, high quality, and timely wind and temperature observations from aircraft surveillance data (Mode-S EHS)**

– We comparison with AMDAR data to the results section

– We included Vector RMS statistics in the results section

– We explain in more detail how the seperate steps for temperature corrections

– We revised the conclusion and added an outlook section

– Lukas Strauss has been added as co-author to give him the credits for inventing the Mach-Indicated Airspeed improvement.

We hope to have answered all remaining questions as good as possible.

Thank you sincerly for your time and effort, Best regards, Siebren de Haan and Co-authors

**Answers**

**General**

ok The EMADDC processing of Mode-S messages into meteorological reports is a major and very useful undertaking. I am a user of the EMADDC reports. In places the jargon (eg ASTERIX CAT48 format) is perhaps too prominent. I have made suggestions about this and minor improvements to the English. The text is sprinkled with more commas than I would use. The statistics all use flight level as the vertical coordinate - the equivalent pressure levels should also be given (at least once).

ok I would like to see some discussion of the wider context, both the impact of Mode-S on NWP and the future of Mode-S and what might replace it. Piecing information together from messages designed for another purpose is not how one would design a meteorological observing system.
*we included some words on this in the conclusion*

ok Also, from my perspective, reports every 4-seconds are overkill. I would hope that, in the longer term it would be replaced by a a better designed aircraft reporting system that provides high resolution data in a single report without the need for heading corrections, rederivation of temperature from Mach number etc.

ok I would like to see some discussion of any moves in that direction, timescale etc and whether it needs a directive from the EU to ensure that such a system becomes widely used over Europe and perhaps elsewhere.

**Detailed**

ok Title: I would suggest 'timely' in place of 'quickly available' and possibly moving 'high volume' before 'high quality' - it is the number of reports that really sets Mode-S apart from other aircraft data sources. What does 'infrastructure' mean here? 'using Mode-S EHS messages' might be better.

ok L1 'Temperature and wind observations from aircraft are regarded of major importance' I suggest 'Wind and temperature . . . ' - the winds are more important.

ok L3 'converts it' - 'converts them' ('data' is a plural noun)

ok L4 'this data' - 'these data'

ok L5 'To acquire' - 'To produce'? 'the data is' - 'the data are'

ok L13 'for example its height, and velocity' 'its' should be 'their' or can be omitted.

ok L15 'aircraft observations form the backbone of the global observing system' 'the backbone' is a bit too stong - 'an important part'?

ok L19 '01/2020 2020', just '2020'

ok L22 'The last decade' - 'Over the last decade'

ok L25,26 'intended heading, airspeed etc.' - I think 'intended' should be deleted (they are reporting actual heading and airspeed)

ok L28 'the most of Mode-S' - 'most Mode-S'

ok L32,33 'observations performed by dedicated aircraft . . . (AMDAR)' - 'observations from AMDAR aircraft' (the acronym was introduced earlier).

ok L35 'ECMWF-IFS' explain acronym (perhaps just ECMWF, need not mention IFS?)

ok L49,50 suggestion: 'not mandatory; fewer than 5% of aircraft respond to such interrogation requests (Strajnar, 2012) and few countries actively interrogate this register.'

ok L51 2.1 Mode-S EHS Interrogation I think parts of this section could be rewritten more concisely.

ok L59 'shall be applied' - 'are applied'

ok L75-78 'Data can be of ASTERIX CAT48 format, which is mono-radar data . . . ' I struggle a bit with the jargon and whether it is useful for me and other users to know. It might be better to put the jargon in brackets, perhaps (assuming that I have understood correctly): 'Data can be from a single radar (in ASTERIRIX CAT48 format) or multiple radars (in CAT62 format, tyically sampled at 4 second intervals; the Mach number is at lower resolution giving derived temperature of lower quality).'

ok L78,79 'For this . . . MUAC to develop a solution.' Perhaps delete the first sentence and replace the second with 'EMADDC is working with EURO-CONTROL MUAC to develop a solution that provides temperature data with consistently good quality.' Also 'MUAC' explain acronym

ok L107 'information of' - 'information on'

ok L112 'Measurements fulfilling one of these checks are discarded' 'failing' better than 'fulfilling'

ok L115 'Output control is necessary to obtain good quality observations.' Please provide details (or possibly a reference) of the quality control applied. Also some indication of the proportion of 'bad' data remaining (1% or 0.1% or whatever), all observing systems have some gross errors. I have recently become aware of some spikes - wind speeds much higher than in the forecast - what might be causing these?
*have no clue; might be a decoding issue.*

ok Table 1. I think that the 'frequency' and 'reported accuracy' headings should be swapped, and 'reported precision' might be better. Do you know if values are rounded or truncated when they are reported?

ok L135 'pressure, at low altitude, is less accurate.' Why?

ok L136 'an improved pressure value that' insert 'is calculated' before 'that'

ok L198 '(minimal 15 days)' - '(at least 15 days)'

ok L231 'is outputted' - 'is output' or 'is written'

ok L236 '9.1 Model comparison' There should be some mention of quality control to remove 'bad' observations (radiosondes as well as Mode-S).

ok Table 4. 'flight level' - 'number' heading is missing 'bias' and 'std.dev' misplaced

ok L248 '9.2 Comparison with Radiosondes observations' - 'Radiosonde' (delete final s)

ok L249 'Radiosondes are regarded as the anchor observation for meteorology' delete 'the' (For satellite soundings GNSS-RO are now more important anchor observations than radiosondes.)

ok L250 'with some sites launching also at 06 UTC and 18 UTC' replace 'some' with 'a few'

ok L250,251 'Due to budget optimization, the number of launches per day was decreased to one or two.' Delete? or replace with 'Due to budget restrictions some radiosondes are only launched once a day.' WMO GBON regards two launches per day as standard and most, but not all, European radiosondes follow this pattern.

ok L251 'Aircraft observations are regarded as replacement to collect upper air observations' - 'Aircraft observations are regarded as supplemental upper air observations'

ok L251-251 'Aircraft and observations will never be collocated in both space and time, . . . . avoids the balloon.' Perhaps just 'will never be exactly collocated' and delete the rest of the senteence.

ok L255 'of 50 km' delete 'of'

ok L255 'The table below' - 'Table 5'

ok L257 'show to have' - 'has'

ok L259 'the mean difference between aircraft and balloon is small.' Both Mode-S and radiosondes have slightly stronger mean wind speeds than 'NWP', I assume that this is because the NWP fields are on a ~9 km grid, whereas the observations are closer to point measurements and have a contribution to the kinetic energy from scales unresolved by the model.
*added some words*

ok Table 6. Column headings missing. Caption too brief.

ok L264,265 'derived from Mode-S EHS aircraft observations' 'reports' or 'messages' better than 'observations' here?

ok L269 'this heading correction is unique for each aircraft individually', delete 'individually'

ok L271,272 'although the data is corrected using ECMWF forecast, the data is independent because aforecast lead time of minimal 9 hours is used' ('minimal' - 'at least') This is only partially true, if the forecast model used has a bias then this will be reflected with reduced magnitude as a bias in the 'corrected' observations, Eyre (2016). Because aircraft heading is not related to forecast values it seems unlikely that the heading correction will cause this type of problem. The temperature and airspeed corrections might be susceptible to problems from model biases. This should be mentioned. Eyre, J.R. (2016), Observation bias correction schemes in data assimilation systems: a theoretical study of some of their properties. Q.J.R. Meteorol. Soc., 142: 2284-2291. https://doi.org/10.1002/qj.2819
*we added some words and a reference to the work of John Eyre*

ok L283,284 'The change in declination is . . . close to zero for low latitude regions (middle panel).' It is confusing having deep red for very small values on this panel - would be better just to use blue scales (white for near zero).

ok Figure A1. define 'WMM' or omit. Add note that the contour intervals are different for the three plots.

ok Appendix B. 'number of observation' - 'number of observations'

ok L296 'in casu'?

ok L302 '1207, E.: Commission' - 'European Commission'?

ok L334 'Painting, J. D.: WMO AMDAR Reference Manual, WMO-No.958, WMO, Geneva, 2003.' WMO regards this manual as superseded (although it is still available), see https://community.wmo.int/en/activity-areas/aircraft-based-observations/resources/manuals-and-guides Should the reference be changed? If not is WMO wrong in regarding it as superseded?

---

## Author Comment (AC3)

Dear Reviewer,

We thank you for your comments and suggustions to improve the manuscript. We have taken the oppurtunity to add several things, some wanted by you some by other reviewers. The main changes in the manuscript are:

– We changed the title to "**EMADDC: high volume, high quality, and timely wind and temperature observations from aircraft surveillance data (Mode-S EHS)**

– We comparison with AMDAR data to the results section

– We included Vector RMS statistics in the results section

– We explain in more detail how the seperate steps for temperature corrections

– We revised the conclusion and added an outlook section

– Lukas Strauss has been added as co-author to give him the credits for inventing the Mach-Indicated Airspeed improvement.

We hope to have answered all remaining questions as good as possible.
Thank you sincerly for your time and effort, Best regards, Siebren de Haan and Co-authors

**Answers**

ok  In the abstract a short paragraph is missing where you describe what is new in your paper namely the correction of pressure through an improved heading and air speed resulting in a new Mach number resulting in a better temperature.

ok  There is no reference of Figure 1 and Figure 2 in the text

ok  Equation 1 needs some references

ok  In line 49-50 the last sentence is not clear understandable

ok  In line 58 correct reseceivers

ok  In your description of the Mode-S EHS interrogation it is not clear to me which time stamp the observations get. Is there also a register for that or does the receiving radar gives the observation a time stamp ?
*we added som words on this*

ok  I think you can combine Section 2.2 and section 2.3 to get one larger section about Aircraft Dependent Surveillance data.

ok  In Section 3 it is not clear to me which way you chose operationally to get better resolved Mach numbers

ok  In Section 2.2 you mentioned the possible assimilation of the difference between GNSS height and pressure altitude. Than you assigned a question mark. Is that a speculation of the authors or to you have other information which are ot published ? Can you reformulate that sentence without a question mark.

ok  In line 89 correct conains.

ok  In table 1 there are two Vertical Rates listed. What is the difference between them ?

ok  What is the difference between airspeed and true airspeed ?
*most of the time we mean 'true airspeed' when we say 'airspeed'; this has been changed accordingly*

ok  In section 6.2 you mentioned that a temperature correction is constructed using NWP temperature information but it is never explained further how and at which point you correct temperature by using NWP data.

35   ok  How do you use the corrected pressure further on. Does it replace the ps value in the formula (4) of the Mach number or how do you use Pcor. The whole section 6.2 can be reformulate to make the temperature bias correction clearer.

ok  Does the corrected True airspeed also makes the derived temperature via the Mach number better ?

ok  In section 8 how do you handle the ADS-B data? In the same way as Mode-S EHS data ?

ok  In your result section you talk about flight levels and kilometers or pressure levels. For clarification it would be nice to assign the flight levels also a
40   height in kilometers or a pressure level.

ok  The description "The tables below" is unclear. Better to say Table 3 to 6 or so

ok  Do you use the ECMWF analyses or a first guess (which) for comparison ?

ok  For the wind direction verification do you discard small wind speeds ? This can become important especially near the surface.

ok  Are the biases and errors compared to the ECMWF model comparable to AMDAR temperature and wind comparisons? Are the values in table 3 and
45   4 high or low ? Can you classify the results ?
*see added paragraph*

ok  In Table 3 and 4 the units are missing

ok  In line 252 Table below again

ok  The sentence in line 257 is unclear.

50   ok  Can you further discuss why the Mode-S EHS temperature data are of less quality in the boundary layer beneath 850 hPa
*see added paragraph*

ok  The conclusions do not clearly describe the novelty of the work. Also, the comparison results between Mode-S data and NWP forecasts or radiosondes can be described in more detail

---

## Author Comment (AC4)

Dear Reviewer, Dear Ed,

We thank you for your comments and suggustions to improve the manuscript. We have taken the oppurtunity to add several things, some wanted by you some by other reviewers. The main changes in the manuscript are:

- we changed the title to "**EMADDC: high volume, high quality, and timely wind and temperature observations from aircraft surveillance data (Mode-S EHS)**

- We comparison with AMDAR data to the results section

- We included Vector RMS statistics in the results section

- We explain in more detail how the seperate steps for temperature corrections

- We revised the conclusion and added an outlook section

- Lukas Strauss has been added as co-author to give him the credits for inventing the Mach-Indicated Airspeed improvement.

We hope to have answered all remaining questions as good as possible.
Thank you sincerly for your time and effort, Best regards, Siebren de Haan and Co-authors

**Answers**

This paper has the potential to be a valuable contribution to the literature and I enthusiastically welcome the submission. However, it does leave the reader with some questions which I believe need to be cleared up before submission. Alongside some other smaller corrections and modifications which will make the message of the paper clear. From my understanding the paper contains two novel themes, 1. EMADDC as a source of high quality aircraft atmospheric observations. 2. An improved method for significantly improved temperatures from Mode-S EHS data.

The other parts are incremental, or explaining how methods previously published have been implemented by EMADDC. I have two significant concerns with the paper as it currently stands,

ok The conclusions do not adequately explain what is novel in the paper. If you were to read it without the context of the entire paper you would just write it off as not novel. The story needs to be made clearer throughout with the conclusions rewritten to draw out the importance and novelty of this work.
*we have rewritten the conclusions*

ok I believe there is a processing method which is mentioned but not adequately explained. The reported quality of the temperatures is significantly greater than that reported previously. It is suggested that this is via creating a corrected pressure and recreating the Mach number. This is never explicitly stated and the method to do this is in no way explained. It is hinted at several times in the paper e.g. Line 80 "derive the Mach number from indicated airspeed", Table 2 (6) the stated minimum Mach is below the resolution limit of the reported Mach and CCC – for CCC the contents of the personal correspondence needs to be detailed, or a further reference found. This is fundamentally the crux and key novel development for Mode-S EHS processing. One of the problems seems to be that the novelty is contained in Straus 2020 which is a personal communication, this needs significant expansion.
*we have rewritten the temperature derivation text. we contacted Lukas Strauss and added him as co-author of the importance of his .....*

I have some less significant comments,

ok The abstract does not fully convey the novelty of the paper, it should also include some highlights of the results e.g. improvement of temperature observations.
*we added some text to the abstract*

ok The use of 'Airspeed' throughout to mean (probably) True Airspeed, this should be stated explicitly at the first introduction or 'true airspeed' should be used throughout, else it is ambiguous with Indicated Airspeed, True Airspeed, Calibrated Airspeed or Equivalent Airspeed.

ok You use altitudes, flight levels and pressure levels in hPa. Please choose one and use it consistently, e.g. Line 262 refers to 858 hPa when refereeing to table 5, table 5 only has flight levels.

ok Section 2.3, this seems superfluous to the rest of the paper.

ok Line 81, where ATC combine the messages to create an observation, how is the time stamp of that 'observation' obtained? The aircraft never reports a time. This is followed up on Line 83 where you talk about it for the receiver, but surely the radar must work the same?

ok Table 1 should also include the reported resolution of the variables.

ok Table 1 it is unclear why there are two "Vertical Rates" in the third section and what each section refers to. It would also be good if this table included the reporting resolution. The BDS table refers to BDS numbers which are not discussed in the text, that makes it unclear, especially as I think some of them are ADS-B messages?

ok Table 6 needs column titles.

ok Lines 140 – 143, You need to explain how Pcor is used to make a new Mach number and/or Temperature? This appears to be phenomenologically derived, is there any physical or instrumental reason for it? I think you need to recreate some of the AMDAR plots for Mode-S to demonstrate it's also true. Can this method really deal with the spread at low M values (e.g. when the aircraft are flying slowly).
*we added some words on this*

ok Table 3, have you considered using the components of the wind rather than speed and direction? That may explain the larger wind direction standard deviations at lower altitudes where the winds tend to be lighter.
*it has been considered...*

And some more minor comments,

ok Figures 1 and 2 appear to not be referenced within the text.

ok Line 22 "The last decade..." – the sentence does not make sense.

ok Line 31, I would also include https://doi.org/10.1175/JTECH-D-15-0184.1 in the reference.

ok Line 40, remove "so called"

ok Line 41, the reference being a number makes it unclear.

ok Line 46, add "and broadcast" to the end, as the broadcast messages are also required.

ok Line 48, add "(where available)" after humidity.

ok Line 57, first use of ADS-B.

ok Line58, the sentence from "is not contained . . . ." Is unclear,you should also reference https://doi.org/10.1175/JTECH-D-15-0184.1 and https://doi.org/10.1029/2010JD

ok Line 66, replace "could be used in DA" with "contains atmospheric information" or similar and reference https://doi.org/10.1175/JTECH-D-14-00192.1

ok Line 77, what is the typical resolution of the Mach number in CAT62?

ok Lines 85 and 86 need rewording as they're unclear.

ok Line 88, your wording of "static pressure or pressure altitude" is somewhat difficult, the pressure altitude is never measured but calculated from the static pressure.

ok Eqn 1, requires referencing.

ok The use of both static and ambient temperature, whilst the meaning is consistent it is probably worth picking one to use consistently, potentially highlighting that it also means the other once in the paper. You may also want to state that this is what NWP/Forecasters expect.

ok Line 105, "arrival of information" – it is unclear as to where this arrives.

ok Line 116, over what time period was the standard deviation calculated?

ok Table 2, how can '6 Mach number smaller than 0.001' be true when the reporting resolution of Mach number is 0.04?

ok Line 132, does it not also result in a correction to the Airspeed?

ok Line 135, what evidence is there that pressure is less accurate at low altitudes? This needs to be referenced, or explained and evidenced.

ok L148 the WMM you are using is quite old, why? My assumption is that your heading corrections will deal with that. IIRC the Met Office uses the IGRF which tries to model the current fields and is updated every year.

ok Line 152, there's a Mirza paper that you should reference. https://doi.org/10.1002/qj.2864

ok Line 185-187, I'm a little confused, do you or do you not use the WMM tables? I'd be really interested to see o-b maps per aircraft.
*yes we are using the world magnetic model*

ok Line 204, could you estimate the order and size of each of the corrections, and reference to the current method of correction?
*tas correction is of the order of -1 to 1 m/s*

ok Line 208, are the ADS-B data decoded and stored in the same way where it's available? *This section is improved to clarify how data is handled for either receivers or radars*

ok Line 215, do you no longer do any interpolation between data points? You don't mention it here but it seemed like it was a significant part of your earlier papers processing methods and based on some recent data discussions it appeared you were still doing it. *We are not doing any interpolation on observations but for several corrections (e.g, TAS and temperature) we determine a mean value of a parameter and use this in the correction.*

ok Line 219, that's interesting. /emphUnfortunately, this was incorrect as the rolling window proved not to be performant. Hence, a simplier linear regression is applied. Futher work on 3.0 will improve this further

ok Lines 226-229, Have you compared these? Primary radar position is worse than ADS-B so why not do something clever in combining? *We have done some investigation into this in the past and concluded that both types of position have pros and cons. E.g, ads-b position is prone to GPS jamming and decoding issues (in case odd/even pairs are missed) whereas radar positions become less accurate further from a radar. Your comment is valid and future research might look into this and see whether we could blend both positions if available*

ok Line 234, "the tables below" which? Reference them, or edit this introduction sentence as it's a bit confusing.

ok Line 237, replace collected with derived as I suspect many more messages were collected!

ok Lines 240 – 244, is this good, bad or indifferent? How do those values compare to other measurement methods?

ok Line 247, do you have any explanations for this?
*not really*

ok Line 252, reference precisely which table.

ok Table 5, is this only the matched observations for EHS – NWP, you should be clear on this point.

ok Line 256, what are the average coherence lengths of wind and temperature fields for these?
*don't know*

ok Table 6 needs column titles.

ok Kube 284, middle panel doesn't seem to agree with the figure, unless I'm misunderstanding and therefore could you edit the words to make it clearer?

110 *done*

---

## Author Comment (AC5)

Dear Reviewer, Dear Ralph,

We thank you for your comments and suggustions to improve the manuscript. We have taken the oppurtunity to add several things, some wanted by you some by other reviewers. The main changes in the manuscript are:

– We changed the title to "**EMADDC: high volume, high quality, and timely wind and temperature observations from aircraft surveillance data (Mode-S EHS)**

– We comparison with AMDAR data to the results section

– We included Vector RMS statistics in the results section

– We explain in more detail how the seperate steps for temperature corrections

– We revised the conclusion and added an outlook section

– Lukas Strauss has been added as co-author to give him the credits for inventing the Mach-Indicated Airspeed improvement.

We hope to have answered all remaining questions as good as possible.

Thank you sincerly for your time and effort, Best regards, Siebren de Haan and Co-authors

**Answers**

Specific Comments:

ok As I read the paper, I kept thinking that the authors might have submitted an earlier version than intended. At the end of line 66, there is a '(?)' in the end of the sentence. What does that mean? The sentence is also conjecture and probably should be eliminated.
*this was erroneous citation.*

ok Throughout the text, there are many instances where statistics that could be quantified are instead replaced by vague adverbs of adjectives, such as the word 'frequently' in the same line. Similarly, in line 53, the words 'not all' should be quantified. As it stands, it could mean that as few as 1% or well over 50% of radars would not meet one of the 2 conditions described in the sentence. Also, Table columns are incorrectly labelled and variables in some of the equations are not clearly defined. Numerous spelling and grammar errors also need to be corrected throughout the paper.
*we improved the paper on this....*

ok Lines 56-59: With the large volumes of Mode-S observations available, how much information does the inclusion of questionably encoded reports add to the volume of reports from more reliable transmissions? Please indicate how much these reports might degrade the overall quality of the derived data sets.

ok *data handling section has been changed* Line 60: It would be very useful to list the transmitted parameters that are more important in deriving each of the meteorological parameters early in the paper. E.g., it would be helpful for the reader to know ahead of time which observed parameters affect temperature derivations.

ok *data handling section has been changed* Lines 73-86: This section describes at least 3 different means in which Mach number that are used at EMADDC. Please explain to the end user how they can know which of the three options were used for in deriving meteorological data from each aircraft and how that choice might affect the quality of the reports and how much difference each of the 3 methods makes.

ok Line 89: Do the Mode-S reports include GPS horizontal position reports as well as altitude? This sentence implies that they do not.

ok Equation 1: It would seem more logical to identify the dynamic pressure at pd instead of qt. Also, the variable in equations 1 and 2 need to be defined in the text.
*we rather stick to $q_t$*

ok Section 4.2: Since this is not relevant to Mode-S observations, this section is unnecessary. Also, the equation, if used, should be written so it is solved as T=, not Ti=.

ok Table 1: This table is incomplete and incorrect in places. Frequency and units are missing for position, even though an accuracy was given in their 2022 paper. The labels of the frequency and reporting accuracy columns are also reversed. Also, although time was listed as a coded parameter in the authors 2022 paper, it is not listed here, nor are the precision of the reported value. This needs to be clarified, since the 2022 paper lists a choice of 2 values (1 s or 1 ms, where 'ms' is undefined). Which is used in your current system? If both are used, what impact does that difference have on derived meteorological variables? Also, no discussion is presented in this or previous papers about how the onboard reports are 'binned' into their reporting precision intervals. Specifically, were the reports simply truncated was software included to determine if the reports were within +/- ½ of the precision interval on either side of the reported value. This information is essential to determine if biases have been introduced in the data compression process.

ok Lines 104-109: Nowhere in the paper are the common frequencies used for the various parameters used in deriving meteorological information specified. This is especially hard for a reader to guess since the frequency is listed as a range. In their 2011 paper, the authors indicated that a 15 (or 60) second averaging (or linear fit) of Mach number and air speed was necessary to improve derived temperature calculations. That statement is not repeated in this paper. Has this changed? If so, say so and explain why. If averaging is used as part of the calculations, then only 1 derived parameter should be reported during the entire averaging period to avoid correlated errors between successive corrections and the reporting frequency should be adjusted to reflect that change. This need to be clarified and well documented. Also, please show which of the two parameters (Mach number or air speed) benefited more for using the linear fit smoothing process? Also, and probably most importantly, is the question of whether the corrections applied to both parameters in the linear smoothing process are correlated or uncorrelated in instances where the method improved the derived temperatures. (For reference, investigation of Mode-S wind speed that I have done using a small random sample of data provided ECMWF shows observation-to-observations wind speed changes frequently approaching +/-2 m/s between successive 20 second reports, even after applying a 3 sigma QC filter. This variability could have major consequences on the quality of more instantaneous temperature derivations.)
*we are now using a time window of 20 seconds for averaging the temperature observations. Wind observations are instantaneous when a corrections are known*

ok Lines 115-118: Having looked carefully at a substantial amount of Mode-S derived meteorological reports, I recommend that error bounds of 2 standard deviations be used instead of 3. This more conservative approach is especially justified give the large volume of Mode-S reports and will reduce the data volume by no more than a few %.
*we will take your observation into account in the upcoming version*

ok Table 2: Please describe how and why these limits were chosen, especially for test 7. Also, if Mach number reporting accuracy is .004, why is .001 used in test 6.
*Mach<0.001 is equal to saying Macch==0*

ok Lines 120-143: This section of the paper concerns me most for numbers of reasons. First, and possibly most significant, is that the fact that the speed of sound (and therefore Mach number) is affected by atmospheric moisture. The 2022 paper explicitly states that the effects of moisture are ignored. Although the effect of moisture is indeed small at upper cruise levels, impacts in the bottom several hundred hPa can be significant, increasing the speed of sound in the moist, less dense environments by over 2 m/s. This can in turn affect temperature derivations by as much as a degree and would result in the derived temperatures being more like virtual temperature than a sensible temperature. This system shortcoming must be recognized. In addition, this large of a change in reported Mach number could shift the transmitted Mach number by one or two of the 0.004 reporting precision increments, which could lead to errors in reported Mach number of up to .008, which could impact temperature calculations even further.
*the section on temperature derivation has been rewritten*

ok Line 129-132: As with all other corrections applied in this paper, please give a typical magnitude and range of values for these corrections. In this case, how much does the static pressure correction affect both Mach number and subsequent derived temperatures?
*see text and newly added figures*

ok Lines 134-143: This paragraph is quite confusing. The first sentence states that NWP is being used to correct temperatures, but no explanation is made of what NWP information is used or how it affects the correction. It implies that corrections are needed for each aircraft individually, but no explanation of the reason for this is documented. A correction formula (formula 5) is then presented based on a new set of static pressures apparently derived from NWP fields. The 2022 paper is then references for more details, but I could not find any there. Instead, the paper describes a method of correcting

AMDAR temperature observations, not Mode-S temperature derivations. An explanation of the derivation of the coefficients in (5) is essential for this technique to be reproduced by others, along with examples of the magnitudes of the correction that are applied as a function of altitude. As a side point, one possible explanation of the effectiveness of the technique is that, since the NWP height fields on pressure surfaces are derived using virtual temperatures, this correction could be unknowingly accounting for the effects of water vapor. This needs much further discussion.

*the section on temperature derivation has been rewritten*

ok Lines 201-205: Although the section on correction for true vs. magnetic north is detailed and seems sound, the statement in lines 201-205 not well documented. No printed reference is given for the 'true air speed bias correction' that EMADDC uses, including no indication of the magnitude of the corrections. It also seems to assume that most of the wind errors are in speed and that wind directions cannot be corrected. Is this true? If so, it should be stated directly. Again, without information about how this correction was formulated, the work cannot be duplicated by others. Finally, if a future physical correction method depends on an already corrected temperature, how might the 2 corrections interact?

*we added a figure and words on the airspeed correction*

ok Section 9.1-9.2: Line 239 refers to the quality of Mode-S wind data using parameters u, v and wind speed. This list, however, fails to include possibly the best measurement of overall wind observations quality, that being the Vector RMS (VRMS), which accounts for both wind speed and direction errors. Although wind speed fits were similar using both NWP and Radiosondes as comparison standards, approximations of the Vector RMS derived from the u and v fits with radiosondes produce VRMS values closer to 3.25 m/s. Comparisons of reports between AMDAR and radiosonde data over the US show wind speed fits less than 2.0 m/s and VRMS values of about 2.5 m/s throughout the depth of the troposphere. TAMDAR reports were substantially worse, with speed fits ranging from 3-4 m/s and VRMS values of 4.5-5.5 m/s. I recommend that you expand your references to include the US intercomparisons, expand your statistics to include VRMS and make specific reference to the lower quality of the individual Mode-S reports when compared to previous AMDAR evaluations. That said, I believe that if the Mode-S reports were amalgamated over periods of 5-7 minutes (the typical time between AMDAR flight level reports), much of the small-scale noise that I have observed will be removed the statistics would improve substantially. Please make the labeling of column headers in Tables 3-6 consistent, more clearly explain the meaning of 'all data' and 'whitelisted and unique' in the caption. Also, please make the layers in Tables 3 and 4 and in Tables 5 and 6 consistent so that the 2 sets of results can be compared directly.

*We added the parameter V RMSE to our statistical environment and the results are shown in the manuscript*

ok I was happy to see that the authors used both NWP and radiosondes in their evaluations. As stated earlier, reliable reports taken by individual aircraft during ascent and descent could be extremely useful for operational bench forecasts, especially in land areas without radiosonde coverage and impending hazardous weather. Finally, although the authors have gone through great lengths to in efforts to derive temperature information from the Mach number of air speed observations available through Mode-S, many fewer temperature derivations were made than the number of wind reports that were made. This reasons for these differences need to be explained (and understood) more clearly in the text, including the methods by which nearly 35% of the derived temperatures were rejected.

*we also added a long time series of AMDAR-EHS comparison to the manuscript*

---

## Author Response (AR2)

**Reviewer 1**

**General comments:**

The paper describes a system that provides wind and temperature data based on aircraft measurements. The European Meteorological Aircraft Derived Data Center (EMADDC) system processes Mode-S Enhanced Surveillance data which is transmitted based on interrogation by ATC radars. Sophisticated methods to minimize errors caused for example by deviations of magnetic heading or truncation errors are introduced. The provision of dense and accurate fields of wind and temperature data in the airspace is of high relevance for aviation meteorology and numerical weather prediction. While the methods to minimize errors and the respective mathematics is quite impressive, there is still space for improving readability of the manuscript. Overall, the paper is a little exhausting to read. One simple means to improve readability could be to revisit the placement of paragraphs. But it would certainly be good to invest some time to improve the writing flow of the text in general. Section 8 is partly not very carefully written. Can you please outline the advantage of EMADDC against AMDAR data? Why do we also need EMADDC? Is it the measurement coverage or frequency? The accuracy of both methods appears similar.

*response* The advantage of Mode-S EHS is that is available (almost) everywhere aircraft go (in Europe). The costs of receiving data are limited, in most area's the data can be obtained through ATC-partners. The disadvantage is the the temperature is not good for heights below 850hPa. We need a good mix of both Mode-S EHS and AMDAR. The text is adapted likewise.

**Specific comments:**

The sizes of the labels of several figures need to be increased.

*response* done

The flow charts in Figs. 1, 2 and 7 are not very nice graphically. Fonts are very small with respect to the space which is not used. The arrows partly overlap words and look like in improvised sketches.

resonse: New flow charts are made

Table 1: Flight levels are usually described in hundreds of feet. Here, possibly a term like flight altitude may be more appropriate?

*response* adjusted

Table 1: Does the reported precision correspond to one stand deviation? How are the values for the precision estimated?

*response* the reported precsision is how many digits are used to describe the parameter value.

Table 1: The global height-keeping performance by ICAO (ICAO, Manual on a 300 m (1000 ft) Vertical Separation Minimum Between FL 290 and FL 410 Inclusive, International Civil Aviation Organization, Doc 9574, AN/934, 3rd edition, 62 pages, 2012) specifies height-keeping errors beyond 90 m (300 ft) in magnitude to less than 0.002 probability. Assuming a Gaussian distribution,

this would yield substantially higher values for one standard deviation than the 25 ft reported here. Maybe for the paper one should state that different error estimates are around and the estimates used here may be connected to substantial uncertainty? Please comment.

*response* the term reported precissions is ambiguous

Table 2: It would be interesting to know whether these values just indicate errors or whether they also exclude phases of flight (like curved flight or approach phase) that would not allow deriving data of good quality or that are not of interest.

*response* looking into any correlation between phase od flight and these flags is an interesring idea and might be looked into in the future. These flags are also being refined in our work for the neW EMADDC 3.0 system

Table 2: The exceedance of roll angles of 2.5° in 16

2.5 degrees of roll is relatively small and can even be measured during turbulenxe. In our opinion, this number is representative. The 2.5 degree limit value is also checked to see if it can be increased in EMADDC 3.0

section 5: The parameters used in the equations should be introduced. It should be checked throughout the paper that all parameters are introduced. Alternatively, a nomenclature could be added.

*response* done

l. 150: The datum y corresponds to the date? According to eq. (11), y is expressed in fractions of a year? Please introduce the definition of datum.

*response* done

Fig. 3 is not easy to understand. The text talks about corrections within the year 2023. However, the y-axis extends over 3 years? The legend denotes years? Please add the unit. The legend only shows values in the past which makes sense to me as the values of the declination table of an aircraft may be outdated. However, the lower plot shows also corrections pointing several decades into the future. Such corrections probably don't make much sense. Shouldn't they be disabled or sorted out accordingly?

*response* done and text has been added

section 5.3: It would be helpful not only to derive mathematically why the heading correction is not sensitive to errors of the numerical wind prediction. Could you please briefly explain in words why this is the case.

*response* text has been added (see 5.3)

l. 210: Numerical weather prediction models may exhibit not only random error but systematic biases in wind speed for certain regions, seasons or altitudes and combinations of these. Isn't there a risk that the correction of magnetic heading in parts corrects deficiencies of the NWP model?

*response* we added some words on this in section "Numerical Weather Prediction model Comparison"

l. 218: "wind vector from the model" does this refer to the NWP model? (There are also other passages where the term model is used and it is not fully clear which model this should be)

*response* added NWP

l. 223: "Care must be taken when model based bias correction are applied because model biases themselves might bias the corrected observation." How can this be done?

*response* we are looking into this

Fig. 4: One may argue whether deviations on the order of 0.1 m/s need to be corrected. Or is this done mainly to get better accuracies for temperature?

*response* correct

Fig. 5: According to the text the error is a variance while in the legend it is a standard deviation.

*response* error and standard deviation ....

Caption Fig. 6: "Red line depicts the mean difference between model and observation (solid line) and standard deviation of the difference (dashed line)." This is not easy to understand. I understand that the bias is a solid line and the STDEV is dashed. Why is the colour red stressed?

*response* this was a typo and is coreected

According table 2 data is not used under certain conditions at FLs below 50. Is this because at these low altitudes temperature errors appear increased in Figs. 5 and 6? Can one explain why temperature errors are increasing at low altitudes and why the bias changes so drastically from negative to positive values at these heights?

*response* the reason is that att low altitudes, aircraft are flying slower, both in Mach as well as airspeed and the truncation of both will introduce larger errors a these heights

Couldn't find a description of Figure 8.

resonse: added a reference

l. 337: "Table 3 also shows the statistics of wind and temperature ..." There is no temperature data in table 3.

*response* corrected

Table 3 provides wind direction errors in the unit m/s!?

*response* corrected

l. 350: Again, the mentioned temperature data is not in the table. Caption of table 5 should also mention the NWP data.

*response* corrected

l. 354: "... has mostly a standard deviation lower .." add mostly, as the statement is not always true

*response* added

l. 357: the model resolution does not correspond to its grid size?

*response* changed simpler wording

Caption of Fig. 9: "Left panel shows the mean differences of temperature ..." add temperature ... "the right panel shows the velocity vector RMS with respect to height" add velocity. Unit of height is missing. The value of 3386 appears not correct.

*response* The value of "3386" is correct, it has been checked

l. 368: the curves rather look like a flight level resolution of 25 has been applied than a 25 ft resolution?

*response* corrected, this was a typo

section 8.3: It should be mentioned that AMDAR is based on the same sensors and the differences between AMDAR and EHS arise from the way the data is processed while of course the difference in location and time also contribute to the deviations.

*response* words have been added

tables 3 -6, Fig. 9 and text: What is labeled as EHS is probably the Mode-S EHS data processed by EMADDC and not the raw EHS data. To avoid misinterpretations EHS should be replaced by EMADDC wherever appropriate.

*response* done

l. 360 and 396: "Comparison with radiosonde observations showed good quality with respect to temperature when the observation is above 850hPa." There is no evidence shown of this claim. In table 6 this height range does not occur explicitly and the errors above FL 400 are bigger than those below FL 100.

*response* you are right we changed the 850hPa to 700hPa and added some words on the increasing standard deviation with height

l. 414: please briefly introduce the meaning of superobbing, as this is not a very common term

*response* we added some words on this

**Reviewer 2**

**General**

This is a useful and interesting manuscript. I want to see the final version include statements to the effect: "Despite the care taken there are still residual errors from the heading correction." and "In the long term it should be better to replace the use of Mode-S derived data with direct reporting of meteorological variables (e.g. via ADS-B)." If the authors don't agree with these statements they should be included as comments from a reviewer.

Response : we added these valuable remarks in the conclusions

From Eyre (2016) and my own experience I take the view that any correction method is imperfect and has residual errors. Learning more about Mode-S, such as occasional errors in interpreting BDS messages or positions, as well as the need for corrections, convinces me that direct meteorological reporting would be better. The fact that EMADDC processing works as well as it does is testament to the hard work and ingenuity applied. As to the question of why the Mode-S and AMDAR wind comparisons to the ECMWF forecast fields are very similar I would suggest that this is partly due to the whitelisting procedure and to the air-speed correction. If similar procedures were applied to AMDAR data I would expect the AMDAR statistics to improve slightly. The residual errors from the heading correction have the advantage that (from most points of view) they are random and hence averaging over 10 or more reports from different aircraft in a small area the errors will largely cancel out. In a thought experiment an aircraft with a residual heading error of 5 degrees (say) travelling from A to B will have

a fairly consistent wind direction error, but as it travels back from B to A the direction error will be opposite. This is assuming that wind speeds are small compared to the air-speed. Some work could be done to firm up these ideas.

*response* at present research on errors is undertaken and a publication will be prepared soon

**Detailed comments**

L83 'Contrary for local receivers' - 'However for local receivers'

*response* Done

L84 'Data received from ATC radars or trackers is also processed ...' - 'also' slightly confusing Is 'tracker' the same as 'local receiver'? 'All data have quality control and filtering applied.' would be clearer

*response* wording changed

L157 'The correction method uses the assumption that the correction is determined by a geomagnetic reference table for a certain datum (or epoch) ...' Why not check this assumption by talking to Airbus, Boeing or others in the aviation industry?

*response* we have made several attenpt to verify this wirh manufactures to no avail unfortinately. Experience and knowledge within our team lead to this assumption however.

Figure 3. Needs clearer explanation. Are the units in years in both parts of the figure? 'maximum heading correction datum' - could be better expressed/explained NB. It is good to see some information on the heading correction applied - but the tails and the noise reinforce the idea that there must be residual errors.

*response* we improved on the wording

Figure 4. I would expect the NWP winds to be representative of a larger area and thus (I think) to have fewer values near the extremes of the distribution. Is this what you see? (I am struggling to fully understand the figure.)

*response* The figure is based on in total 12 hours of data and therefor extremes (when present) are not averaged out

L243 'is of worse accuracy' - 'has worse accuracy'

*response* done

L256 'dashed blue line' - not dashed!

*response* done

'8.2 Comparison with Radiosonde Observations' Are the radiosonde data being used at the launch position (as used to be standard) or at the position they have drifted to?

*response* we used the drift

414 'to apply this most efficient' - 'to apply this most efficiently'

*response* done

Acknowledgements: Add ECMWF for their forecast data?

*response* of course!

**List of major changes**

1. The text has been improved using the revierwers comments and by careful reading itself

2. all flowcharts have been redrawn and look better now (we think)

3. variables aused in equations are described

4. Figure captions have been checked and adapted

5. we added some words on "direct measurements" vs "indirect and corrected measurements

---

## Author Response (AR3)

EMADDC: high volume, high quality, and timely wind and temperature observations from aircraft surveillance data (Mode-S EHS)

We thank Bruce for reviewing!

Detailed comments

line 27. "Dedicated aircraft" - "Some commercial aircraft" would be better *done*

40 "preferred above" - "preferred to" *done*

40 "Since corrections" - "Corrections" (or combine the two sentences: "paramters, since corrections") *done*

191 "Using the found datum for an aircraft" - "Using this value for the aircraft" (also change datum to value line 189) *done*

200 "are not regular flying" - "are not flying regularly" *done*

201 "high correction datum values" - "large corrections" *done*

202 "datum corrections values" - "correction values" *done*

206 "The corrections for these aircraft are invalidated and hence not used." - I suspect you mean that the winds from these aircraft are not used. It could be read that they are used with zero correction. *done* Some words have been added

223 "many of observations" - "many observations" *done*

237 "few tenth meter per second" - "few tenths of a meter per second" *done*

320-321 "outliers are identified using linear regression over individual aircraft's 30-second time windows"

A colleague pointed out that looking at a time-series of hours for an individual aircraft there are some suspect points (outliers). Some checking over a time period longer than 30-seconds should be considered. *we will consider this when working on the next version of heading and temperature correction*

354 "historic NWP" - "past NWP" *done*

355 "operational available" - "current" (matches past better) *done*

360 "These values are all within the acceptable range for use in for example data assimilation." It is worth mentioning the WMO Rolling Review of Requirements, see https://space.oscar.wmo.int/observingrequirements (I haven't done any comparison of the two) *done* we added some words

370 "as anchor observation" - "as anchor observations" *done*

370 "are generally launched at the main synoptic hours 00 UTC . . . " they are usually launched 40 or 45 minutes earlier. Perhaps "provide profiles centred on the main synoptic hours 00 UTC . . . " *done* we added some words

376 "has mostly" - "mostly has" *done*

379 "Reason" - "The reason" *done*

381 "radiosonde soundings" *done*

Figure 9. I suggest removing the legend from the first and last panels as they partially obscure the lines. Add vertical zero line to the first panel. Caption "middle pane" - "middle panel" (or use 'pane' for all three) *done*

458 "National Meteorological Hydrological Institutes" - "National Hydrometeorological Institutes" (usual term) *done*

Ruijgrok, G. J. J.: Elements of Airplane Performance, Delftse University Pers, 1990. "Delft University Press" There is a second edition 2009 https://repository.tudelft.nl/record/uuid:9ff3c7e0-e221-4d8f-bf9e-e144d3cacdc1 *not done* thanks for the link, next time we will use this reference